# Supersymmetry in the time domain and its applications in optics

Carlos García-Meca [1,2]*, Andrés Macho Ortiz [1,2]* & Roberto Llorente Sáez [1]

Supersymmetry is a conjectured symmetry between bosons and fermions aiming at solving fundamental questions in string and quantum field theory. Its subsequent application to quantum mechanics led to a ground-breaking analysis and design machinery, later fruitfully extrapolated to photonics. In all cases, the algebraic transformations of quantum-mechanical supersymmetry were conceived in the space realm. Here, we demonstrate that Maxwell's equations, as well as the acoustic and elastic wave equations, also possess an underlying supersymmetry in the time domain. We explore the consequences of this property in the field of optics, obtaining a simple analytic relation between the scattering coefficients of numerous time-varying systems, and uncovering a wide class of reflectionless, three dimensional, all-dielectric, isotropic, omnidirectional, polarisation-independent, non-complex media. Temporal supersymmetry is also shown to arise in dispersive media supporting temporal bound states, which allows engineering their momentum spectra and dispersive properties. These unprecedented features may enable the creation of novel reconfigurable devices, including invisible materials, frequency shifters, isolators, and pulse-shape transformers.

[1] Nanophotonics Technology Centre, Universitat Politècnica de València, Valencia 46022, Spain. [2] These authors contributed equally: Carlos García-Meca, Andrés Macho Ortiz *email: cargarm2@ntc.upv.es; amachor@ntc.upv.es

Supersymmetry (SUSY) was conceived as a fundamental symmetry of string and quantum field theory that could allow the unification of all physical interactions of the universe[1–5]. Subsequently, the field of supersymmetric quantum mechanics (SUSYQM) was created with the aim of solving essential questions about SUSY via a non-relativistic model[6]. Basically, the simplest version of SUSYQM considers two different one-dimensional (1D) systems governed by the eigenvalue equations:

$$\hat{H}_{1,2}\psi^{(1,2)}(x) = \Omega^{(1,2)}\psi^{(1,2)}(x), \qquad (1)$$

where $\hat{H}_{1,2} = -\alpha d^2/dx^2 + V_{1,2}(x)$ are the Hamiltonians (with $\alpha > 0$), $V_{1,2}$ the potentials, and $\Omega^{(1,2)}$ the eigenvalues. The central idea is to define an auxiliary function $W$ (known as the superpotential) and two SUSY operators $\hat{A}^{\pm} = \mp\sqrt{\alpha}d/dx + W(x)$ such that $\hat{H}_1 = \hat{A}^{+}\hat{A}^{-}$. The second Hamiltonian is then constructed by inverting the operator order, i.e., $\hat{H}_2 = \hat{A}^{-}\hat{A}^{+}$. The potential of SUSYQM resides in the fact that $\hat{H}_1$ and $\hat{H}_2$ have the same scattering properties and eigenvalue spectrum. As a consequence, although SUSY has not been experimentally observed in nature[7], SUSYQM has become a revolutionary mathematical tool in itself, enabling the explanation of intriguing aspects of quantum mechanics (such as the existence of non-trivial reflectionless potentials and of very different systems with the same energy spectrum), uncovering new analytically-solvable potentials, and offering a simple and systematic way to construct infinite families of isospectral quantum-mechanical systems[6].

Interestingly, Eq. (1) also governs the dynamics of other physical phenomena. This is case of electromagnetic waves in certain kinds of media, which enables a direct extrapolation of the SUSYQM formalism to optics[8,9]. As a result, the basic ideas of this theory have recently led to pioneering photonic structures[9–14].

Being a 1D theory, one could ask whether a temporal supersymmetry might exist for time-varying potentials. Nevertheless, to our knowledge, the SUSYQM formalism has never been applied in the time domain, whether in QM, optics, or any other field (SUSY quantum field theory is a multidimensional spacetime theory, but the formalism is considerably different and more complex than that of SUSYQM). This is probably due to the fact that the vast majority of 1D SUSY work has been developed within the realm of QM, and the time derivative in Schrödinger's equation is of first order, preventing a similar decomposition to that of Eq. (1) in the time domain (time-dependent potentials have been considered in SUSYQM, but also using SUSY operators based on first-order spatial derivatives[15,16], making it impossible to exploit the potential of the standard spatial SUSY (S-SUSY) factorisation in the time domain). On the other hand, only a few works deal with optical SUSY, all focused on S-SUSY. Remarkably, however, the fact that the temporal derivative in the electromagnetic, acoustic, and elastic wave equations is of second order may enable a temporal version of SUSYQM, which has been overlooked so far. This would extend the foundations and unique properties of SUSYQM to the time domain, adding an unprecedented degree of understanding and control over time-varying systems in various fields of physics, and opening the door to a myriad of new applications. Actually, time-varying optical systems are becoming crucial in a broad range of scenarios, including optical modulation[17], isolation and non-reciprocity[18,19], all-optical signal processing[20,21], quantum information[22], and reconfigurable photonics[23,24]. Likewise, temporal modulations enable new possibilities for the manipulation of sound and mechanical oscillations[25–27].

Here, it is shown that Maxwell's equations indeed possess an underlying time-domain supersymmetry (T-SUSY) for any non-dispersive optical system characterised by a refractive index of the

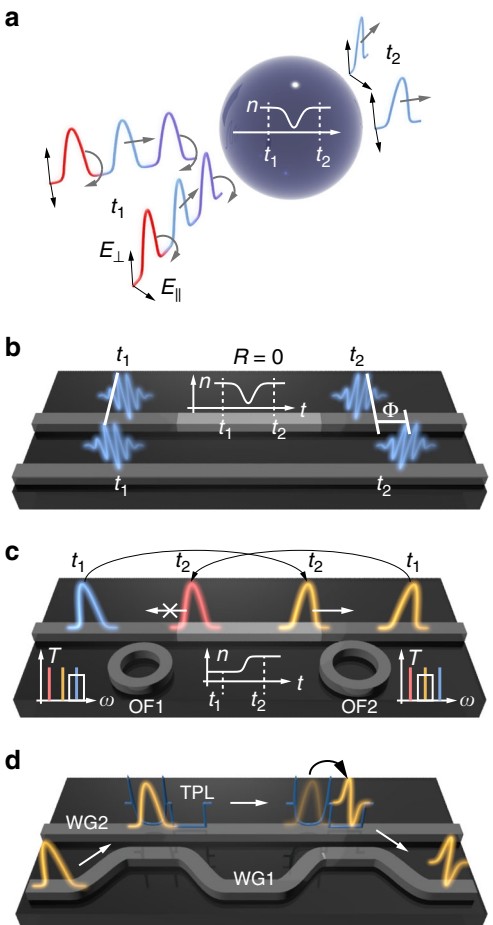

**Fig. 1 Potential T-SUSY applications. a** Omnidirectional, isotropic, polarisation-independent, all-dielectric (all-magnetic), 3D time-varying material that is invisible in a given (reconfigurable) spectral region, allowing the generation of frequency-selective transparent temporal windows. **b** Perfect phase shifter: a short region in a waveguide (lighter grey) with a fast T-SUSY time-varying index inducing a dynamically-reconfigurable frequency-independent reflectionless phase shift $\Phi$ over an optical pulse. **c** Optical isolator: another T-SUSY time-varying index shifts the frequency of a right-propagating pulse (blue to yellow here), which can traverse the optical filters OF1 and OF2. Any left-propagating pulse is reflected at OF1 or OF2, protecting a left-side source from external reflections. **d** Reconfigurable pulse-shape transformer: an input pulse propagating along waveguide WG1 is spatially coupled to a T-SUSY temporal photonic lantern (TPL, a moving index perturbation) running along waveguide WG2. The pulse excites another TPL mode with a different desired shape, coupled back to WG1.

form:

$$n(\mathbf{r},t) = n_S(\mathbf{r})n_T(t) = \sqrt{\varepsilon_S(\mathbf{r})\mu_S(\mathbf{r})}\sqrt{\varepsilon_T(t)\mu_T(t)}, \qquad (2)$$

where $\varepsilon_r(\mathbf{r},t) = \varepsilon_S(\mathbf{r})\varepsilon_T(t)$ is the medium relative permittivity and $\mu_r(\mathbf{r},t) = \mu_S(\mathbf{r})\mu_T(t)$ its relative permeability, with similar results for acoustic and elastic waves (T-SUSY can also be found in dispersive systems, as discussed below, and in anisotropic and nonlocal media, as discussed in Supplementary Note 1). In the following, the T-SUSY formalism is developed for the field of optics, analysing both the continuous and discrete-spectrum cases, and illustrating its potential through different applications (sketched in Fig. 1). Finally, the extension of T-SUSY to transmission line theory, acoustics and elasticity is discussed, assessing the experimental opportunities offered by current technological platforms.

## Results

**Continuous spectrum.** Consider a linear, isotropic, heterogeneous, time-varying non-dispersive medium with $n_T^2(t) = \varepsilon_T(t)$. Applying separation of variables in the electric flux density $D(\mathbf{r}, t) = \phi(\mathbf{r})\psi(t)$ of Maxwell's equations, we find that $\psi(t)$ exactly obeys the Helmholtz's equation:

$$\left(\frac{d^2}{dt^2} + \omega^2 N^2(t)\right)\psi(t) = 0, \tag{3}$$

where $N^2(t) := n_-^2/n_T^2(t)$, $n_- := n_T(t \to -\infty)$, and $\omega$ is the angular frequency of the field at $t \to -\infty$. For a polychromatic wave, the total field is given by the superposition of the solutions to Eq. (3) for each spectral component (value of $\omega$). Equation (3) is also obtained for $n_T^2(t) = \mu_T(t)$ and even for general materials with $n_T^2(t) = \varepsilon_T(t)\mu_T(t)$ (in which case, $\varepsilon_T(t)$ and/or $\mu_T(t)$ must vary slowly in time), see Supplementary Note 1. Equation (3) exactly matches Eq. (1) taking $\alpha = 1$, relabelling $x \to t$, and identifying $\Omega - V(t) \equiv \omega^2 N^2(t)$. Using the eigenvalue $\Omega$ as a degree of freedom, this will allow us to apply 1D SUSY in the time domain, with two fundamental noteworthy features: (1) T-SUSY is exact for both all-dielectric and all-magnetic indices $n_T$; (2) T-SUSY is completely uncoupled from space. Hence, it is valid for all polarisations, all propagation directions and any 3D spatial medium dependence $n_S^2(\mathbf{r}) = \varepsilon_S(\mathbf{r})\mu_S(\mathbf{r})$. This means that we can generate T-SUSY partners of devices such as waveguides or structures with any desired 3D scattering response while keeping the spatial properties of interest (e.g., ability of guiding or reflecting/refracting the fields in a specific way for each direction and polarisation; see Supplementary Note 1 for an example involving an ideal polariser, which shows that the spatial response associated with a time-invariant refractive index is preserved by its T-SUSY partner for all polarisations simultaneously). Contrarily, 1D SUSYQM is, by definition, only valid for 1D spatial variations, and only for a specific polarisation in the optical case[9–11]. Furthermore, T-SUSY can be used to study temporal scattering in systems with continuous spectra, as well as time-varying systems supporting discrete-spectrum bound states. In both cases, its application is not as straightforward as that of S-SUSY.

First, unlike in S-SUSY, to develop T-SUSY for wave scattering, the concept of negative frequencies is essential. This comes from the differences between spatial and temporal scattering, exemplified in Fig. 2 with a simple model having one spatial dimension. As is well known, when a plane wave traverses a localised spatial variation in a time-invariant medium, there appear reflected and transmitted waves of the same frequency (photon energy), with the wave number (photon momentum) of the incident ($k_-$), reflected ($k_R$) and transmitted ($k_+$) waves fulfilling the Snell's relations: $k_R = -k_-$ and $k_-/n_- = k_+/n_+$, resulting from spatial symmetry breaking (Fig. 2a, Supplementary Movie 1). Less known is the fact that, when a wave propagates through a homogeneous medium, reflections also appear under a localised time variation (Fig. 2c, Supplementary Movie 2). In this case, since only time symmetry is broken, momentum is conserved and photon energy changes, with the frequency of the incident ($\omega_- = \omega$), reflected ($\omega_R$) and transmitted ($\omega_+$) waves obeying the relations[28] $\omega_R = -\omega_+$ and $n_+\omega_+ = n_-\omega_-$. That is, light can exchange energy with the medium. Notably, the frequencies of the reflected and transmitted waves have opposite signs. Although the physical meaning of negative-frequency waves is striking and controversial[29,30], mathematically, the Hermiticity of the fields in $k$–$\omega$ space allows reinterpreting a negative-frequency wave as a counter-propagating positive-frequency one, leading to the standard use of only-positive frequencies. However, the introduction of negative frequencies in this work is not a mere convention. It is a mathematical tool that enables the analysis of temporal

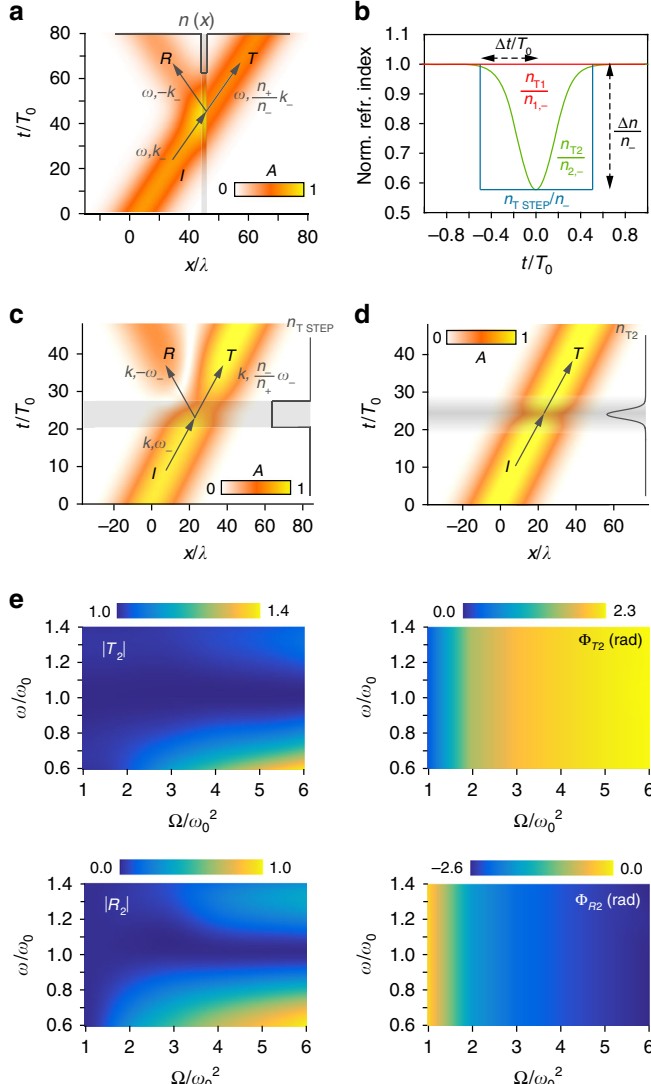

**Fig. 2 Reflectionless all-dielectric (all-magnetic) T-SUSY time-varying optical media. a** Example of spatial reflection for an optical beam going through a time-invariant spatial step-index medium (see Supplementary Movie 1). **b** Normalised refractive index profiles $n_T(t)/n_-$ of homogeneous media characterised by a temporal step-index $n_{TSTEP}(t)$ (always reflective), a constant index $n_{T1}(t)$ (non-reflective), and its T-SUSY partner $n_{T2}(t)$ (non-reflective), with $\Omega = 2\omega_0^2$. **c, d** Pulse propagation evolution at $\omega = \omega_0$ through the media with index profiles $n_{TSTEP}(t)$ in (**c**) and $n_{T2}(t)$ in (**d**) (numerically calculated for the case $n_- = 2$). Here, $\lambda = \lambda_0/n_-$ and $\lambda_0 = 2\pi c_0/\omega_0$. In contrast to (**c**), the result in (**d**) demonstrates the reflectionless nature of $n_{T2}$. See Supplementary Movies 2 and 3. **e** Scattering coefficients $T_2 = |T_2|e^{i\Phi_{T_2}}$ and $R_2 = |R_2|e^{i\Phi_{R_2}}$ of $n_{T2}$ as a function of $\omega/\omega_0$ and $\Omega/\omega_0^2$.

scattering and, more importantly, a necessary ingredient to relate the reflection and transmission coefficients of T-SUSY index profiles, which otherwise cannot be decoupled (Supplementary Note 2). Concretely, for a given system with a temporal index $n_{T1}$, T-SUSY provides a systematic way of generating a superpartner, whose index is (see Supplementary Note 2):

$$n_{T2}(t) = \frac{n_{2,-}}{\sqrt{\frac{n_{1,-}^2}{n_{T1}^2(t)} - \frac{2}{\omega^2}W'(t)}}, \tag{4}$$

where $n_{1,2,\pm} := n_{T1,2}(t \to \pm\infty)$ is assumed to be constant. As seen,

the prescription for $n_{T2}(t)$ depends on $\omega$. Since it would be challenging to realise such a frequency-dependent index in practice, we consider a more realistic and typical scenario in which the medium has the same temporal variation for all relevant frequencies of the electromagnetic field. This variation is taken to be the one that makes $n_{T1}(t)$ and $n_{T2}(t)$ supersymmetric at a frequency $\omega_0$, which will be a free design parameter (it can be, e.g., the central frequency of the spectral band of interest, or any other reference frequency providing the desired response). That is, we suppress the frequency dependence by fixing $\omega$ to $\omega_0$ in Eq. (4). Consequently, the two systems will be guaranteed to be exact T-SUSY partners at $\omega_0$, while they may exhibit different properties at other frequencies. The same situation is found in S-SUSY[9,12]. Nevertheless, it can be shown that the T-SUSY connection generally holds almost exactly in a broad frequency band, which is a typical feature of supersymmetric optical systems[10,12]. This can be verified by calculating the solution to the wave equation associated with $n_{T2}$ at $\omega \neq \omega_0$ (Supplementary Note 1), from which the medium spectral response can be obtained (an example is given below for a reflectionless system, see Fig. 2). It is worth mentioning that any system with a given relative index variation $n_T(T_0 t_N)/n_-$, with $t_N := t/T_0$ and $T_0 := 2\pi/\omega_0$, will have the same normalised solution $\psi(T_0 t_N)$, and therefore the same scattering properties. Thus, for the sake of generality, the normalised variables $t_N$ and $n_T/n_-$ are used whenever possible.

Note that Eq. (3) admits asymptotic solutions for $n_{T1,2}$ in the form of the following incident, reflected and transmitted plane waves: $\psi_I^{(1,2)}(t \to -\infty) = e^{i\omega_0 t}$, $\psi_R^{(1,2)}(t \to \infty) = R_{1,2} e^{-iN_+ \omega_0 t}$ and $\psi_T^{(1,2)}(t \to \infty) = T_{1,2} e^{iN_+ \omega_0 t}$, where $N_+ := n_{1,-}/n_{1,+} = n_{2,-}/n_{2,+}$. The combined use of negative frequencies and T-SUSY then relates the reflection and transmission coefficients of both media as (Supplementary Note 2):

$$R_1 = \frac{W_+ + iN_+ \omega_0}{W_- - i\omega_0} R_2, \quad T_1 = \frac{W_+ - iN_+ \omega_0}{W_- - i\omega_0} T_2, \tag{5}$$

where $W_\pm := W(t \to \pm\infty)$ and $W$ is obtained by solving the first-order Riccati equation $V_{1,2}(t) = W^2(t) \mp W'(t)$, with $V_{1,2}(t) = \Omega - \omega_0^2 N_{1,2}^2(t)$. Two important consequences can be inferred from Eq. (5): (1) it enables us to directly obtain the reflection and transmission coefficients of numerous intricate time-varying optical systems (in general, of any T-SUSY partner of a known-response system), circumventing the resolution of Maxwell's equations; (2) $|R_1| = |R_2|$ and $|T_1| = |T_2|$ (as a direct consequence of the fact that $n_{T1}$ and $n_{T2}$ share the same eigenvalue $\Omega$). As a result, T-SUSY will enable us to straightforwardly construct families of time-varying media having the same scattering intensity as another one (with the desired spatial variation and polarisation response), implementable over the same ($n_{2,-} = n_{1,-}$) or a different ($n_{2,-} \neq n_{1,-}$) index background, bringing about a variety of applications.

As an example, consider the simplest case: a constant refractive index $n_1(\mathbf{r}, t) = n_{1,-}$. Its T-SUSY partner is $n_2(\mathbf{r}, t) = n_{T2}(t) = n_{2,-}[1 + 2(\Omega/\omega_0^2 - 1)\text{sech}^2(\sqrt{\Omega - \omega_0^2} t)]^{-1/2}$ (Fig. 2b, Supplementary Movie 3). The free parameters $n_{2,-}$ and $\Omega$ allow tailoring the asymptotic value of $n_{T2}$, as well as its maximal index variation $\Delta n$ and temporal width $\Delta t$ (defined as the required time interval to obtain $\Delta n$, see Supplementary Note 3). Since $n_{T1}$ is constant, $R_1 = 0$. Therefore, $n_{T2}$ will also be reflectionless as demonstrated in Fig. 2d for a quasi-monochromatic pulse. From our previous discussion, $n_{T2}$ represents a new class of all-dielectric (all-magnetic), omnidirectional, isotropic, polarisation-independent, and invisible 3D media with real positive ($>1$) permittivity (permeability). No known spatially-varying material

possesses all these features simultaneously, including transformation media[31], complex-parameter materials[32], and S-SUSY media[10]. The only previously reported time-varying reflectionless media required to concurrently induce temporal modulations in the permittivity and permeability[33]. Our T-SUSY proposal is totally different, since it is valid for all-dielectric $n_2^2(\mathbf{r}, t) = \varepsilon_S(\mathbf{r})\varepsilon_T(t)$ and all-magnetic materials $n_2^2(\mathbf{r}, t) = \mu_S(\mathbf{r})\mu_T(t)$, see Supplementary Note 1. The former are particularly important, as implementing temporal permittivity modulations is extremely easier than implementing permeability ones.

As discussed above, another general feature of T-SUSY is that it is only exact for the design frequency $\omega = \omega_0$. Therefore, $n_2$ will be invisible ($R_2 = 0$, $|T_2| = 1$) for all directions and polarisations at $\omega_0$, while a reflected wave will appear at other frequencies. The response of $n_2$ at $\omega \neq \omega_0$ will be characterised by the value of $R_2$ and $T_2$ as a function of frequency, which can be rigorously obtained by solving numerically Supplementary Equation 8. The result for the present example is depicted in Fig. 2e, which not only confirms the expected reflectionlessness of $n_2$ at $\omega = \omega_0$, but also demonstrates that this property is almost preserved in a wide spectral band. For example, for $\Omega = 2\omega_0^2$ (which corresponds to a rapidly-varying index with $\Delta t/T_0 = 0.6$), the system has a reflectance $|R_2|^2 < 10^{-3}$ in a fractional bandwidth $F = \Delta\omega/\omega_0 = 30\%$. Moreover, the spectral span for which $n_2$ is almost invisible ($R_2 \approx 0$, $|T_2| \approx 1$) can also be tailored through $\Omega$, enabling us to generate custom-made transparent temporal windows within $n_2$ only for desired spectral bands (Figs. 1a and 2). Concretely, $F$ decreases as $\Omega$ increases. For instance, $|R_2|^2 < 10^{-3}$ in a fractional bandwidth $F > 55\%$ for $\Omega = 1.5\omega_0^2$, while $|R_2|^2 < 10^{-3}$ in a bandwidth $F = 8\%$ for $\Omega = 6\omega_0^2$. Remarkably, out of the invisible band, light is (partially) retroreflected along the input path, in contrast to spatial retroreflectors, in which the reflected path is parallel to, but different from, the input one[34].

Moreover, $n_2$ exhibits a singular feature: the phases of the reflection and transmission coefficients ($R_2 = |R_2|e^{i\Phi_{R_2}}$, $T_2 = |T_2|e^{i\Phi_{T_2}}$) have a frequency-independent spectral response, adjustable via $\Omega$ (see Fig. 2e and Supplementary Fig. 3). As a result, $n_2$ implements a perfect dynamically-reconfigurable phase shifter with the property of being reflectionless, polarisation- and frequency-independent, and of having a short response time ($\Delta t < 5\pi/\omega_0$) in the generation of any phase shift $\in [0, \pi]$ (allowing a significant reduction of the device length), blazing a trail for designing ideal ultra-compact optical modulators (Fig. 1b, Supplementary Note 3). In contrast, optical-path-based phase shifters usually demand slowly-varying index modulations (and therefore devices with a higher length) to have a negligible reflection, and are intrinsically frequency-dependent[35,36]. Contrariwise, the proposed T-SUSY device may induce the same phase shift over different spectral channels, which could be of great utility in, e.g., frequency combs and wavelength-division multiplexing technology. Likewise, pulse shaping operations can also be implemented by taking advantage of the nonlinear spectral response of $\Phi_{T2}$ (see Supplementary Note 3). Additional T-SUSY media emerge from a constant index (being therefore reflectionless), such as the hyperbolic Rosen-Morse II (HRMII) potential (Fig. 3a, and Supplementary Fig. 8), which provides a free design control over $\Delta n$ for a fixed response time ($\Delta t \sim 20\pi/\omega_0$), allowing a technology-oriented tuning of the index excursion.

It is worth mentioning that the origin of reflectionless time-varying systems may be explained in terms of the absence of Stokes phenomenon (here related to the asymptotic behaviour of the solution to Eq. (3) in the limit $\omega \to \infty$)[37], which also explains the origin of transitionless quantum systems[38]. The latter can be obtained through the so-called transitionless tracking algorithm, which, given a non-adiabatic Hamiltonian $\hat{H}_0(t)$, generates a

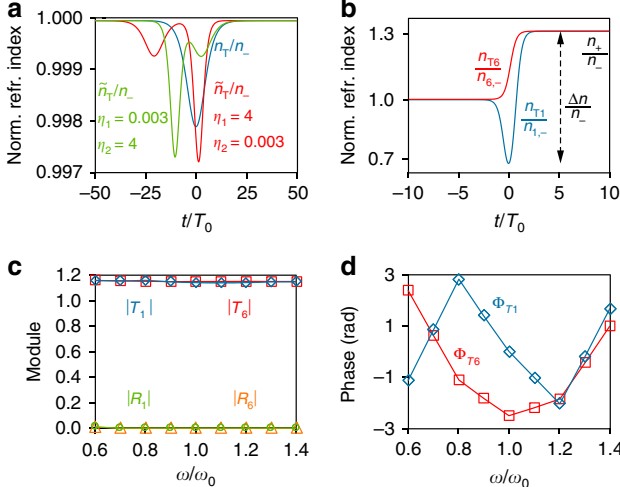

**Fig. 3 T-SUSY reflectionless isospectral media and frequency converter.**
**a** Two members of the 2-parameter isospectral family $\tilde{n}_T(t; \eta_1, \eta_2)$ of the HRMII index $n_T(t)$. Numerical calculations show that, in all the analysed cases with $n_- = 2$, $R(\eta_1, \eta_2) = 0$ and $T(\eta_1, \eta_2) = e^{i0.23}$ (Supplementary Note 3). **b** Normalised supersymmetric refractive index profile $n_{T1}(t; a_1)/n_{1,-}(a_1)$ of Eq. (6) with $a_1 = 40$, $B = a_1^2/10$, $\alpha = 10$, $\omega_0 = 38$ rad·s$^{-1}$. The sixth order normalised index of its corresponding SI chain $n_{T6}(t; a_1)/n_{6,-}(a_1)$ is also depicted. Both media induce a reflectionless frequency down-conversion in any incident optical signal (see Supplementary Movie 4). **c**, **d** Module (**c**) and phase (**d**) of the scattering coefficients of $n_{T1}(t; a_1)$ and $n_{T6}(t; a_1)$ as a function of frequency (calculated for $n_{1,-}(a_1) = n_{6,-}(a_1) = 2$). The phase of $R_1$ and $R_6$ cannot be estimated due to the non-reflecting behaviour of $n_{T1}(t; a_1)$ and $n_{T6}(t; a_1)$ in an extremely large optical bandwidth. No reflected wave is observed in the numerical simulation when propagating wide-band optical pulses through these time-varying media.

second Hamiltonian $\hat{H}(t)$ (or a family of them) that drives the instantaneous eigenstates of $\hat{H}_0(t)$ exactly without transitions between them. Nevertheless, note that transitionlessness and reflectionlessness are not equivalent in general. To see this, consider the case in which $\hat{H}_0$ does not depend on time, and is hence transitionless (in fact, $\hat{H} = \hat{H}_0$ from equations 2.7 and 2.9 of ref. [38]). If $\hat{H}_0$ is not reflectionless, $\hat{H}$ will not be reflectionless either, showing that transitionlessness does not necessarily imply reflectionlessness. Although it might be possible to obtain time-varying reflectionless systems with some version of this kind of method, we have not found any example in the literature (also note that the transitionless quantum driving algorithm[38] does not apply to the temporal Helmholtz's equation). In any case, the transitionless method and T-SUSY are different formalisms. For instance, if the original Hamiltonian is the one associated with a constant potential, the transitionless algorithm just returns the same system, while T-SUSY can generate infinite non-trivial reflectionless systems from a constant potential.

Actually, the results we have presented so far can be extended via different T-SUSY variants. Firstly, isospectral T-SUSY deformations provide a route to obtain $m$-parameter index families $\tilde{n}_T(t; \eta_1, \ldots, \eta_m)$ with exactly the same scattering properties in module and phase as another medium $n_T$ (Supplementary Note 2). As an example, Fig. 3a shows a reflectionless two-parameter family of the HRMII index. Secondly, for some $n_{T1}$ profiles, shape invariance (SI) allows us to construct T-SUSY index chains $\{n_{Tk}(t; a_1)\}_{k=1}^m$ satisfying the relations $n_{Tm}(t; a_1) \propto n_{T1}(t; a_m)$, $R_m(a_1) = R_1(a_m)$, $T_m(a_1) = T_1(a_m)$ and $a_m = f(a_{m-1}) = (f \circ f)(a_{m-2}) = (f \circ f \circ \ldots \circ f)(a_1)$, with $f$ a real function. Therefore, we can straightforwardly analyse or design

the temporal scattering properties of a large number of time-varying media. To illustrate the benefits of SI, consider the following variation of the HRMII index ($\alpha$ is a real parameter):

$$n_{T1}(t; a_1) = \frac{n_{1,-}(a_1)}{\sqrt{1 - \frac{2B}{\omega_0^2} + \frac{a_1(a_1 + \alpha)}{\omega_0^2}\operatorname{sech}^2(\alpha t) - \frac{2B}{\omega_0^2}\tanh(\alpha t)}}, \quad (6)$$

which is also reflectionless in a wide spectral band (Fig. 3c and Supplementary Fig. 11). Since $n_{1,-} \neq n_{1,+}$, the system performs a frequency down-conversion with $\omega_+ = (n_{1,-}/n_{1,+})\omega_-$, where $n_{1,-}/n_{1,+}$ can be engineered via the design parameters $\omega_0$ and $B$. A device exhibiting all these properties has many potential applications. Unfortunately, the exotic shape (reaching values below $n_{1,-}$) and large maximal excursion of $n_{T1}(t; a_1)$ hampers its experimental implementation. T-SUSY can overcome this drawback by using SI. Specifically, Eq. (6) satisfies the SI condition with $a_m = a_1 - (m - 1)\alpha$, allowing us to generate different index profiles with the same reflectionless band as $n_{T1}(t; a_1)$ (see Fig. 3b, c and Supplementary Movie 4). Taking $m = 6$, we find an index $n_{T6}(t; a_1) = [n_{6,-}(a_1)/n_{1,-}(a_6)]n_{T1}(t; a_6)$ with a considerably smoother time variation and a significantly lower $\Delta n$. As illustrated in Fig. 1c, $n_{T6}(t; a_1)$ can be used to build a polarisation-independent optical isolator with an ideally unlimited bandwidth (unlike previous time- and spacetime-modulated isolators and frequency converters[18,39,40], which, in addition, usually involve complicated non-omnidirectional implementations), difficult to achieve by other means (Supplementary Note 3 includes more details on this device and additional SI examples). Reflectionless frequency converters can also be designed via transformation optics, but their implementation requires extremely complex spacetime-varying bianisotropic materials[41].

**Discrete spectrum.** Let us now discuss the discrete-spectrum case. This scenario naturally arises in optical S-SUSY. Specifically, when applying 1D SUSYQM to a spatial dimension normal to the propagation direction, the propagation constant enters the wave equation as an effective energy, which is quantised by the eigenvalue problem[9,10]. However, since time is unidimensional, there is no possible quantity playing the role of an energy in T-SUSY, leading to free-particle systems. Outstandingly, a discrete-spectrum version of T-SUSY can be developed for time-varying dispersive media. To this end, we require the concept of temporal waveguide (TWG): two adjacent temporal index boundaries (allowed to move at a speed $v_B$) defining a position-dependent temporal index window, which can confine optical pulses by temporal total internal reflection[42–44]. TWGs can be created by inducing a perturbation $\Delta n_{\text{eff}}(t - z/v_B)$ of the effective index $n_{\text{eff}}$ of a given mode in a spatial waveguide. In a co-moving reference frame, the complex envelope of the electric field associated with a TWG can be written as[43,44] $A(z, \tau) = \sum_n \psi_n(\tau) e^{i(\Delta\beta_1/\beta_2)\tau} e^{iK_n z}$. It is then shown that a TWG supports temporal bound states $\psi_n$ ($n = 0, 1, 2, \ldots$) fulfilling the discrete-spectrum eigenvalue equation:

$$\left(-\frac{d^2}{d\tau^2} + 2\frac{\beta_B(\tau)}{\beta_2}\right)\psi_n(\tau) = \left(2\frac{K_n}{\beta_2} + \frac{\Delta\beta_1^2}{\beta_2^2}\right)\psi_n(\tau), \quad (7)$$

where $\tau := t - z/v_B$, $\beta_1$ and $\beta_2$ are the inverse group velocity and group-velocity dispersion constant of the perturbed spatial mode, $\Delta\beta_1 = \beta_1 - 1/v_B$, $\beta_B(\tau) = k_0\Delta n_{\text{eff}}(\tau)$, $k_0 = \omega_0/c_0$, and $\omega_0$ is the optical carrier angular frequency. We can apply T-SUSY to Eq. (7), as it matches Eq. (1) for $\alpha = 1$, $x \to \tau$, $V(\tau) := 2\beta_B(\tau)/\beta_2$ and $\Omega_n \equiv 2K_n/\beta_2 + \Delta\beta_1^2/\beta_2^2$, expanding the TWG landscape and its potential applications.

As an example, consider an analytically-solvable TWG with a step temporal perturbation $\beta_{B1}$. Its unbroken T-SUSY partner is $\beta_{B2}(\tau) = \beta_{B1}(\tau) - \beta_2(\ln \psi_0^{(1)}(\tau))''$, where $\psi_0^{(1)}$ is the ground state

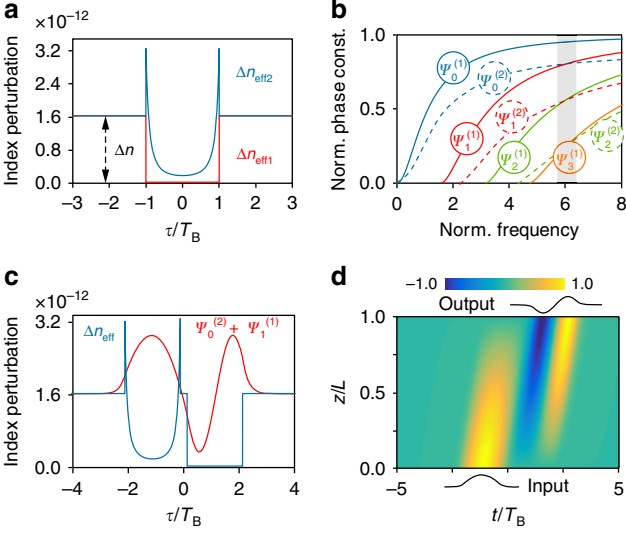

**Fig. 4 Supersymmetric TWGs and TPLs. a** Temporal index perturbations $\Delta n_{eff1}$ and $\Delta n_{eff2}$ of a step-index TWG ($2T_B = 660$ ps, $|\Delta\beta_1| = 10^{-3}$ ps·m$^{-1}$, $\beta_2 = 0.06$ ps$^2$·m$^{-1}$) and its T-SUSY partner. **b** Normalised dispersion diagram $b-\nu$ of the temporal bound states $\psi_n^{(1)}$ and $\psi_n^{(2)}$ of both TWGs, where $b$ is the normalised phase constant, defined for each $n$-th order mode as $b_n := 1 - K_n/\Delta\beta - \Delta\beta_1^2/(2\Delta\beta\beta_2)$, and $\nu$ is the normalised frequency, with $\nu^2 := 2T_B^2\Delta\beta/\beta_2$ (here, $\nu = 6$). $\psi_n^{(2)}$ exhibits a lower slope (is less dispersive) than $\psi_{n+1}^{(1)}$. The grey area is the phase-matching bandwidth, the interval $\Delta\nu$ where $\Delta b \leq 0.2$ between SUSY bound states. **c** Index perturbation of a TPL (blue) constructed from two T-SUSY TWGs with a time separation $T_B/4$ ($2T_B = 660$ ps), and its temporal supermode (red), generated from the perfect phase-matching between the states $\psi_0^{(2)}$ and $\psi_1^{(1)}$. **d** Pulse-shape transformation resulting from the energy transfer between $\psi_0^{(2)}$ and $\psi_1^{(1)}$ in the TPL (Supplementary Movie 5).

(fundamental mode) of $\beta_{B1}$ (Fig. 4a, Supplementary Note 4). From T-SUSY theory, both TWGs have the same energy spectrum. Moreover, $\hat{A}^-$ ($\hat{A}^+$) maps each state $\psi_n^{(1)}$ ($\psi_n^{(2)}$) of $\beta_{B1}$ ($\beta_{B2}$) into a state of $\beta_{B2}$ ($\beta_{B1}$) having the same eigenvalue $\Omega_n$, with the exception of the ground state $\psi_0^{(1)}$, which is annihilated by $\hat{A}^-$ ($\hat{A}^- \psi_0^{(1)} = 0$) and thus has no equal-energy counterpart in $\beta_{B2}$. Particularly, $\psi_n^{(2)} \propto \hat{A}^- \psi_{n+1}^{(1)}$, where $\hat{A}^- := d/d\tau - (\ln\psi_0^{(1)}(\tau))'$. In T-SUSY, energy is related to phase constant, implying that $\psi_0^{(1)}$ is not phase-matched with any $\psi_n^{(2)}$ and that $\psi_{n+1}^{(1)}$ and $\psi_n^{(2)}$ are perfectly phase-matched. This occurs in an extremely large optical bandwidth $\Delta\nu \sim 0.5$ ($\nu \propto 1/\beta_2$ is the normalised frequency), provided that both TWGs are built on a dispersion-flattened spatial waveguide ($d\beta_2/d\omega \approx 0$), see Fig. 4b. Furthermore, Fig. 4b reveals that $\beta_{B2}$ is less dispersive than $\beta_{B1}$, i.e., T-SUSY enables us to engineer the dispersion properties of TWGs.

To further unfold the potential of discrete-spectrum T-SUSY, we propose the concept of temporal photonic lantern (TPL): close-packed serial T-SUSY TWGs moving at the same speed and supporting linear combinations of degenerate temporal bound sates (supermodes)[45]. To verify the TPL concept, we have developed the first version of coupled-mode theory (CMT) for serial TWGs (Supplementary Note 4). Figure 4c shows an example of a TPL supermode. Remarkably, TWGs can carry soliton-like (shape-invariant) optical pulses in a time-varying dispersive medium, with the advantage of enabling arbitrary pulse amplitude, phase and duration, as well as a tuneable propagation speed[43,44]. TPLs extend this ability to a serial combination of modes (each with arbitrary length, amplitude and node number), yielding solitonic supermodes with almost any desired shape.

Achieving the required perfect phase-matching between modes of different order in serial TWGs without T-SUSY typically demands neighbouring TWGs of different width, whilst T-SUSY permits an independent control over this parameter and generally presents a much larger normalised phase-matching bandwidth[12,14], inherently implying a higher tolerance to fluctuations in $T_B$ and $\Delta n_{eff}$. More advanced functionalities emerge by noting that if only one of the TWGs of the TPL is excited, a periodic energy transfer between adjacent TWGs occurs (Fig. 4d, Supplementary Movie 5, Supplementary Note 4). Using two coupled spatial waveguides (WG1, WG2), this effect enables the construction of a pulse-shape transformer with unprecedented versatility and reconfigurable capability (Fig. 1d). Particularly, the final shape of each pulse propagating along WG1 can be dynamically chosen among a large gamut by launching the appropriate TPL over WG2. The proposed T-SUSY TPLs could also find application in optical wavelet transforms, coherent laser control of physicochemical and QM processes, spectrally-selective nonlinear microscopy, and mathematical computing[46–49]. T-SUSY TWG theory can be extended via temporal analogues of SI, broken SUSY and isospectral constructions[6,14].

## Discussion

Overall, these results generalise the foundations of SUSYQM to the time domain, unveiling the temporal supersymmetric nature of Maxwell's equations (which, unlike optical S-SUSY, is a genuine symmetry with no previous direct analogue) and, consequently, leading to the emergence of an entire field of research within physics, as well as to a new photonic design toolbox. Compared with S-SUSY, T-SUSY relaxes the need for controlling the polarisation state of light and the medium spatial index variation, which usually involves complex fabrication steps[9,10]. In addition, as outlined above and as shown in detail in Supplementary Notes 7 and 8, both sound and elastic waves satisfy a temporal Helmholtz equation formally equal to Eq. (3). Hence, T-SUSY can be directly transferred to these fields of physics.

A possible T-SUSY technological difficulty might arise if high temporal index excursions and/or index variations with a very low temporal width are desired (e.g. to achieve a large phase shift in an ultra-compact optical device). Since the response and achievable functionalities of the system depend solely on $n_T(T_0 t_N)/n_-$, the limits of a given technology will be determined by the maximum (minimum) allowed values of $\Delta n/n_-$ ($\Delta t/T_0$). Table 1 summarises the main features of the most suitable existing platforms for the implementation of T-SUSY index modulations.

At optical frequencies, strongly-nonlinear epsilon-near-zero (ENZ) media such as indium tin oxide (ITO) and aluminium-doped zinc oxide (AZO) provide large values of $\Delta n/n_-$ (7.2 and 4.5, respectively, widely exceeding the figures here considered), but are limited by their high $\Delta t/T_0$ values (36 and 33, respectively) and typical linear losses[50–52]. Silicon carbide (SiC) is an interesting alternative material that possesses an ENZ band at a wavelength $\lambda_{ENZ} \approx 10.33$ μm, with a low estimated response time[52–54] $\Delta t/T_0 \approx 5$ and a maximum[51,54,55] $\Delta n/n_- \approx 1$. Hence, modulations such as those depicted in Fig. 3 could be potentially realised with this kind of material. Note that nonlinear ENZ media with smaller values of $\Delta t/T_0$ in combination with low linear losses might emerge with the development of new materials[52].

Phase change materials (PCMs) also support large values of $\Delta n/n_-$ at high frequencies. As an example, germanium antimony telluride (GST) alloys provide variations of $\Delta n \approx 3$ ($\Delta n/n_- \approx 0.9$) with very low losses at $\lambda = 10$ μm and a controllable degree of

**Table 1 Potential T-SUSY implementation platforms.**

| Technology | Frequency ($\omega_0/2\pi$) | Maximum normalised index excursion ($\Delta n/n_-$) | Minimum normalised response time ($\Delta t/T_0$) |
|---|---|---|---|
| Nonlinear ENZ[50-55] (ITO, AZO, SiC[a]) | 30–242 THz | 1–7 | 5–36 |
| PCMs[56-58] (GST) | 30 THz | 0.9 | 12,000 |
| TVTLs[61,62] | 0.4–2.4 GHz | 2.5 | 0.1 |
| Acoustics[25,64,65] | 0.5–1.7 kHz | ≥0.1[b] | Not reported |
| Elasticity[27,66] | ≈5 Hz | 0.4 | 0.3 |

[a]In the case of SiC, the value of $\Delta t/T_0$ has been estimated by considering a similar rise time to that of ITO and AZO, while the value of $\Delta n/n_-$ has been calculated by using the general theory reported in ref. [51] along with the SiC properties reported in refs. [54,55].
[b]The value reported in ref. [25] is $\Delta n/n_- = 0.1$. Considerably higher values can be achieved with other approaches, including tuneable metamaterials allowing near-zero mass density values (and thus large normalised index variations)[64] or electronically-controlled active acoustic metamaterials reconfigurable in real time[65].

crystallisation enabling a gradual index variation[56,57]. Currently, the high typical response time of PCMs ($\Delta t \geq 400$ ps, $\Delta t/T_0 \approx 12,000$)[58] would be the main drawback of this technology for the implementation of T-SUSY modulations. Nevertheless, the femtosecond phase transitions achieved in some experiments via non-thermal mechanisms[59] may lead to faster PCM technology at optical frequencies in the future.

In low-frequency electromagnetism, time-varying transmission lines (TVTLs) based on a dynamically-tuneable distributed capacitance constitute an ideal experimental platform to test and exploit all the proposed T-SUSY concepts, as their effective dielectric properties can be largely tuned in space and time. In particular, the capacitance of commercial varactor diodes can be modified by a factor of 12 (corresponding to $\Delta n/n_- \approx 2.5$)[60], while their response time can be as low as $\Delta t/T_0 \approx 0.1$ (see Supplementary Note 6)[61,62].

In acoustics, a temporal index modulation can be induced through variations of the medium mass density (Supplementary Note 7). There exist a number of strategies to achieve strong changes of this parameter, including the use of magnetoacoustic crystals[63], acoustic metamaterials[64], or acoustic systems with moving parts[25]. Notably, there exist more sophisticated electronically-controlled active materials whose local acoustic parameters can be changed almost arbitrarily in real time[65].

Moreover, T-SUSY systems can be implemented in elastic beams with a time-varying stiffness $D_T$, which allow temporal modulation amplitudes as high as $\Delta D_T/D_- = 1.4$ (corresponding to $\Delta n/n_- \approx 0.4$) and normalised response times as low as $\Delta t/T_0 = 0.3$ (see Supplementary Note 8)[27,66].

Finally, note that extremely low index perturbations suffice to create T-SUSY TWGs, technologically feasible in standard optical fibres and waveguides via travelling-wave electro-optic phase modulators or the cross-phase modulation effect[30,43,44].

## Methods

**Numerical calculations.** The temporal scattering problem (Figs. 2 and 3) has been simulated by solving numerically Supplementary Equation 8 with the commercial software COMSOL Multiphysics. In order to guarantee a low computational time, we have taken $\omega_0 = 38$ rad·s$^{-1}$ and $c_0 = 1$ m·s$^{-1}$. Nevertheless, the results of Figs. 2 and 3 are valid for any value of $\omega_0$ and $c_0$ (see Supplementary Note 3 for more details). On the other hand, the discrete spectrum of the TWGs and the TPL (Fig. 4) has been analysed with the commercial software MATLAB and CST Microwave Studio by using the analogy between the modes of a TWG and those of a dielectric slab waveguide reported in ref. [43]. Finally, the pulse-shape transformation depicted in Fig. 4d has been calculated in MATLAB by solving the CMT reported in Supplementary Note 4 for serial TWGs. See Supplementary Note 5 for more details about the methods employed in Fig. 4.

## Data availability

The data that support the findings of this study are available from the corresponding authors upon reasonable request.

## Code availability

The codes generated during the current study are available from the corresponding authors upon reasonable request.

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

## Acknowledgements
This work was supported by Spanish National Plan projects TEC2015-73581-JIN PHUTURE (AEI/FEDER, UE) and MINECO/FEDER UE XCORE TEC2015-70858-C2-1-R, as well as Generalitat Valenciana Plan project NXTIC AICO/2018/324. A.M.O.'s work was supported by BES-2013-062952 F.P.I. Grant.

## Author contributions
C.G.-M. conceived the idea of temporal SUSY. C.G.-M. and A.M.O. developed the theory, performed the numerical simulations and analysed the data in equal contribution. R.L.S. supervised the work. All authors contributed to write the paper.

## Competing interests
The authors declare no competing interests.
