## [Peer Review File · Nature Communications]

Reviewers' comments:

Reviewer #2 (Remarks to the Author):

In this work the authors consider the temporal analogue of supersymmetric quantum mechanics, applied to Maxwell's equations. Interestingly it is found that, provided the permittivity is chosen in product form $\epsilon_1(x)\epsilon_2(t)$, then a separation of variables yields a one dimensional Helmholtz equation for the time variation of the displacement field, with a separation constant that is related to the initial frequency of a monochromatic wave. This equation can be factorized in exactly the same manner as the 1d Helmholtz equation in terms of a spatial variable. As a result supersymmetric transformations can be applied to find time varying material parameters with identical bound states, and/or identical scattering properties.

Overall I liked this paper, and I am not aware of the core idea appearing anywhere else in the literature. However, I have a number of doubts concerning its current suitability for Nature Communications:

(1) This work does not reference any of the previous work initiated by Berry related to "transitionless quantum driving" (J Phys A 2009). This is closely related to the problem of time varying reflectionless potentials. As Berry puts it, these time varying potentials are

"explained by continuation to complex time, where the complex degeneracies in the transitionless driving fields have a nongeneric structure for which there is no Stokes phenomenon... this is analogous to the explanation of reflectionless potentials"

This statement refers to Berry's earlier work on the non-reflecting potentials (including the Poschl-Teller potential, which is also discussed in the work of the authors) - see "Fake Airy functions and the asymptotics of reflectionlessness" (J Phys a, 1990). However, although there is much follow up work to the 2009 paper, I could not find any work where supersymmetry was applied to Maxwell's equations in the time domain.

(2) Many authors avoid the consideration of time varying optical properties because they are so difficult to realise on the necessary time scales (for e.g. optical frequencies I think any significant variation of ϵ will occur over a timescale such that everything is adiabatic and thus no reflection can be expected). I think to be suitable for publication there needs to be some concrete prospect of realising these temporal profiles experimentally. The idea of using non-linearity is a good one, but this will restrict Δn to small values. Can another higher contrast example be given? Are there examples of such time variations in e.g. acoustics or low frequency EM?

(3) The restriction to a fixed initial frequency ω_0 seems quite severe. I think it is important to spell out in more detail what this restriction implies. For example it is also stated that this can be the 'central frequency' in the non-monochromatic regime, but it was not clear to me whether this was an approximation, and if so under what conditions it holds.

Reviewer #3 (Remarks to the Author):

The manuscript "Optical Supersymmetry in the Time Domain" demonstrate time domain version (T-susy) of optical susy. The authors numerically and analytically show a new paradigm of optical susy in time domain with potential applications such as temporally controlled e.g. phase shifter and a frequency converter.

The paper is well-written and the results and discussion are sound and convincing. However, I would like the two notes below are addressed before publication.

1. Can the authors exemplify a material with a temporal change in refractive index as shown in Fig. 1b inset ($\Delta n = 2 - 1.2 = 0.8$)? if not could they replace figure one with a real material values?

2. Authors show that T-susy equations are uncoupled from space. Accordingly, it is claimed that T-susy is polarization independent and omnidirectional (page 4) which is unlike one dimensional S-susy. (Off course this is true, since there is no direction limitation for time.) As an example in page 6, a constant refractive index material with no reflection is considered. I assume that the material is infinite that has no boundary with free space. Then it is shown that a new class of all-dielectric (all-magnetic), omnidirectional, isotropic, polarization-independent, and transparent 3D media with real positive (>1) permittivity (permeability) can be introduced using T-susy.

Here is the question:

If the material is polarization dependent (consider an ideal a polarizer) that reflects for one polarization and transmit for the corresponding orthogonal polarization, will T-susy be reflectionless, for all polarizations? In other words, if n_{T2} gives reflectionsless class of materials for the same polarizer at one polarization, will the amount of reflection be 100% for the orthogonal polarization with the same n_{T2} , or n_{T2} needs to be changed to obtain 100% reflection in orthogonal polarization. I think authors need to address this in the paper.

Review of the Nature Communications paper NCOMMS-19-13626-T “Optical Supersymmetry in the Time Domain”

Please find below our feedback on the reviewers’ comments. The modifications included in the revised version of the manuscript (R paper) are described after each comment and marked in red in the R paper text.

Feedback from Reviewer #2 Comments

General comments:

In this work the authors consider the temporal analogue of supersymmetric quantum mechanics, applied to Maxwell’s equations. Interestingly it is found that, provided the permittivity is chosen in product form $\epsilon_1(x)\epsilon_2(t)$, then a separation of variables yields a one dimensional Helmholtz equation for the time variation of the displacement field, with a separation constant that is related to the initial frequency of a monochromatic wave. This equation can be factorized in exactly the same manner as the 1D Helmholtz equation in terms of a spatial variable. As a result supersymmetric transformations can be applied to find time-varying material parameters with identical bound states, and/or identical scattering properties.

Overall I liked this paper, and I am not aware of the core idea appearing anywhere else in the literature. However, I have a number of doubts concerning its current suitability for Nature Communications:

The authors appreciate the referee’s positive comments, suggestions, and detailed review of the paper.

1. This work does not reference any of the previous work initiated by Berry related to “transitionless quantum driving” (J Phys A 2009). This is closely related to the problem of time-varying reflectionless potentials. As Berry puts it, these time-varying potentials are “explained by continuation to complex time, where the complex degeneracies in the transitionless driving fields have a non-generic structure for which there is no Stokes phenomenon... this is analogous to the explanation of reflectionless potentials”. This statement refers to Berry’s earlier work on the non-reflecting potentials (including the Poschl-Teller potential, which is also discussed in the work of the authors) - see “Fake Airy functions and the asymptotic of reflectionlessness” (J Phys a, 1990). However, although there is much follow up work to the 2009 paper, I could not find any work where supersymmetry was applied to Maxwell’s equations in the time domain.

We thank the reviewer for bringing these works to our attention. We were not aware of the concept of “transitionless quantum driving”, which we found very interesting. We have carefully studied these papers and, as the reviewer mentions, there are connections between reflectionless and transitionless systems at a certain level. From what we understand, these connections are basically related to a particular method of explaining the origin of reflectionlessness or transitionlessness, as proposed by Berry in [Ber90] and [Ber09], respectively. Specifically, the method consists of approximating the characteristic parameter of the system (e.g., the potential in Schrödinger’s equation) by a local expansion around a zero or pole of its complexified version. Then, the asymptotic expansion of the *solution* to the resulting equation is studied, finding that it terminates at the first term. This implies that, unlike in most systems, no additional contribution is required to fulfill the equation under consideration (no reflected wave in the case of reflectionless systems, no excitation of other states in the case of transitionless systems), due to the absence of Stokes phenomenon.

However, as far as we understand, this connection does not go beyond the similarity in the way of explaining both phenomena. Although one could establish a semantic analogy between transitionlessness and reflectionlessness (e.g. by referring to the coupling from an incident wave to a reflected one as a transition), in general, these two concepts are not equivalent

To see this, consider, for instance, the case of a Hamiltonian \hat{H}_0 that does not depend on time. Then, from Eqs. (2.7) and (2.9) of [Ber09], its transitionless version is $\hat{H} = \hat{H}_0$. This was to be expected, since, being a time-invariant Hamiltonian, \hat{H}_0 is already transitionless and needs not be corrected to obtain a transitionless

system. Therefore, if \hat{H}_0 is not reflectionless, its transitionless version \hat{H} will not be reflectionless either, showing that transitionlessness does not necessarily imply reflectionlessness.

It is reasonable to think that there might exist some version of the transitionless algorithm with which to obtain time-varying reflectionless systems (in particular in the case of the temporal Helmholtz's equation considered in the present work), although, after a thorough examination of the literature, we have not found any example of this.

In any case, we would like to remark that supersymmetry (SUSY) and the transitionless tracking algorithm are different formalisms. In this respect, there is no reference reporting a formal relation between them to our knowledge (nor any previous application of SUSY to Helmholtz's or Maxwell's equations in the time domain).

As a matter of fact, the transitionless algorithm does not relate two systems with the same properties. Instead, given a non-adiabatic Hamiltonian \hat{H}_0 , it finds another Hamiltonian \hat{H} (or a family of them) that drives the instantaneous eigenstates of \hat{H}_0 exactly with no transitions between them. On the contrary, SUSY allows us to generate families of systems with the same properties as another one. This includes the ability of producing systems with the same energy spectrum (except the ground state if desired) and scattering properties (not only for reflectionless systems), or even of obtaining the complete energy spectrum of a given potential – via shape invariance – without knowing the spectrum of its partner (attributes that the transitionless method does not possess) [Coo95]. The ability of SUSY to generate infinite families of reflectionless systems is only a particular application of these qualities.

As another example, in the transitionless method, calculating the correcting term generally requires knowing the spectral properties of the original instantaneous Hamiltonian at all times. This wide knowledge is not required to obtain SUSY partners of a given system.

As a final example, if the original Hamiltonian is the one associated with a constant potential, the transitionless tracking algorithm returns the same system, as discussed above. This is a crucial difference with respect to SUSY, which provides infinite non-trivial reflectionless systems starting from a Hamiltonian with a constant potential.

Following the reviewer's comment, and in order to highlight and clarify the connection between supersymmetric and transitionless systems, we have included Refs. [Ber90] and [Ber09] in the list of references of the revised manuscript (now Refs. 33 and 34), and added the following explanation on page 8:

“It is worth mentioning that the origin of reflectionless time-varying systems may be explained in terms of the absence of Stokes phenomenon (here related to the asymptotic behaviour of the solution to equation (3) in the limit $\omega \rightarrow \infty$)³³, which also explains the origin of transitionless quantum systems³⁴. The latter can be obtained through the so-called transitionless tracking algorithm, which, given a non-adiabatic Hamiltonian $\hat{H}_0(t)$, generates a second Hamiltonian $\hat{H}(t)$ (or a family of them) that drives the instantaneous eigenstates of $\hat{H}_0(t)$ exactly without transitions between them. Nevertheless, note that transitionlessness and reflectionlessness are not equivalent in general. To see this, consider the case in which \hat{H}_0 does not depend on time, and is hence transitionless (in fact, $\hat{H} = \hat{H}_0$ from equations (2.7) and (2.9) of ref.³⁴). If \hat{H}_0 is not reflectionless, \hat{H} will not be reflectionless either, showing that transitionlessness does not necessarily imply reflectionlessness. Although it might be possible to obtain time-varying reflectionless systems with some version of this kind of method, we have not found any example in the literature (also note that the transitionless quantum driving algorithm³⁴ does not apply to the temporal Helmholtz's equation). In any case, the transitionless method and T-SUSY are different formalisms. For instance, if the original Hamiltonian is the one associated with a constant potential, the transitionless algorithm just returns the same system, while T-SUSY can generate infinite non-trivial reflectionless systems from a constant potential.”

References

- [Ber90] M. V. Berry, “Fake Airy functions and the asymptotics of reflectionlessness,” J. Phys. A: Math. Gen. **23**, L243 (1990).
[Ber09] M. V. Berry, “Transitionless quantum driving,” J. Phys. A: Math. Theor. **42**, 365303 (2009).
[Coo95] F. Cooper, A. Khare, and U. Sukhatme, “Supersymmetry and quantum mechanics,” Physics Reports **251**, 267 (1995).

2. Many authors avoid the consideration of time-varying optical properties because they are so difficult to realise on the necessary time scales (e.g. for optical frequencies I think any significant variation of ϵ will occur over a timescale such that everything is adiabatic and thus no reflection can be expected). I think to be suitable for publication there needs to be some concrete prospect of realising these temporal profiles experimentally. The idea of using non-linearity is a good one, but this will restrict Δn to small values. Can another higher index contrast example be given? Are there examples of such time variations in e.g. acoustics or low frequency EM?

The authors thank the referee's comments and suggestions on the importance of the experimental implementation of the proposed T-SUSY systems, which, together with the comments of reviewer #3 along the same line, have stimulated us to considerably improve this aspect in the manuscript. In the following, we address each point separately (subsections 2.1 - 2.5). The corresponding modifications we made to the manuscript and to the Supplementary Information, as well as some final remarks on this issue, can be found in subsections 2.6 and 2.7, respectively.

2.1. Many authors avoid the consideration of time-varying optical properties because they are so difficult to realise on the necessary time scales (e.g. for optical frequencies I think any significant variation of ϵ will occur over a timescale such that everything is adiabatic and thus no reflection can be expected).

We agree with the referee that, at optical frequencies, it is still challenging to experimentally realise strong refractive index temporal variations in a time scale Δt (defined as the required time interval to obtain the maximum index variation Δn) of the same order of magnitude as that of the electromagnetic wave oscillation period T_0 . Nonetheless, recent technological advances are continuously pushing down the ratio $\Delta t/T_0$. As an example, large normalised index variations $\Delta n/n_- \approx 1$ [$n_- := n(t \rightarrow -\infty)$] could be attained in a relatively low normalised response time $\Delta t/T_0 \approx 5$ by stimulating the Kerr nonlinear effect of silicon carbide (SiC) near its epsilon-near-zero (ENZ) wavelength ($\lambda_{\text{ENZ}} \approx 10 \mu\text{m}$) [Res19, Kim16, Che15].

On the other hand, it should be noted that $|R|$ (and all the scattering properties of a system) depends on the variation of n/n_- rather than on that of n alone [Hay16] (this can be seen by noting that the only medium parameter entering the temporal Helmholtz's equation is $N = n_-/n$, see Eq. (3) of the manuscript). Therefore, the reflection induced by a temporal index modulation with a large value of $\Delta n/n_-$ cannot always be ignored, even if Δn is not too high or if $\Delta t/T_0 \gg 1$ (the usual case at optical frequencies). This becomes especially apparent when employing strongly-nonlinear ENZ materials. At optical frequencies, $n_- \approx 10^{-1}$ in this kind of media, while Δn can be as high as 0.7 [Ala16], so reflections might not be negligible. To show this, we have numerically estimated the reflection of three strongly-nonlinear optical ENZ materials: (i) indium tin oxide (ITO) [Ala16] (ii) aluminium doped zinc oxide (AZO) [Cas16], and SiC [Kim16, Che15]. For simplicity, and taking into account that the index variation of these materials is approximately a linear function of time (see e.g. Fig. 2D of [Ala16]), we have modelled the transition from n_- to n_+ by the following profile:¹

$$n(t) = \begin{cases} n_-; & t < 0 \\ n_- + \frac{(n_+ - n_-)}{\Delta t} t; & 0 \leq t \leq \Delta t \\ n_+; & t > \Delta t \end{cases} \quad (\text{r2.1})$$

For ITO, $n_- = 0.1$, $n_+ = 0.82$, and $\Delta t \approx 36T_0$ [Ala16], which results in $|R| \sim 9\%$ ($20 \log(0.09) = -21$ dB of reflected power), while $n_- = 0.1$, $n_+ = 0.55$, and $\Delta t \approx 33T_0$ for AZO [Cas16], yielding $|R| \sim 5\%$ (-26 dB). As for SiC, which, as mentioned above, has a normalised response time $\Delta t/T_0 \approx 5$ [Kim16, Res19], the maximum normalised index variation can be estimated to be $\Delta n/n_- \approx 1$ in its ENZ band [Che15, Leo17, Cas16].² With these parameters, the calculated reflection for SiC is $|R| \sim 10\%$ (-20 dB). Although these values are small, they must be taken into account for some applications, since a reflection amplitude $|R| > 3\%$ (< -30 dB) may directly induce a relevant noise level in lasers [Hir79, Tem86] and, indirectly, in optical communication systems and devices via backscattering events [Hma11, Jia07]. T-SUSY would allow us to reduce this reflection level.

¹ Numerical simulations of the temporal scattering problem have been performed by solving Supplementary Eq. (1.8) with COMSOL Multiphysics.

² We have estimated $\Delta t/T_0$ in SiC by considering a similar rise time to that of the ITO and AZO. In addition, $\Delta n/n_-$ has been estimated in SiC by using the general theory reported in [Cas16] along with the SiC properties reported in [Che15, Leo17].

Finally, it is worth mentioning that, as suggested in [Res19]:

“There might, of course, be a better material platform that is still to be discovered. The ideal material would have the following properties: CMOS compatibility, high degree of crystallinity, large carrier mobility and low linear losses. If possible, this material would have also a tailorable zero-permittivity wavelength, and it would be straightforward to deposit and nanostructure.”

For instance, the emergence of novel strongly-nonlinear ENZ media with a higher λ_{ENZ} than that of the aforementioned materials [Res19] would lower the ratio $\Delta t/T_0$.

References

- [Ala16] M. Z. Alam, I. De Leon, and R. W. Boyd, “Large optical nonlinearity of indium tin oxide in its epsilon-near-zero region,” *Science* **352**, 6287 (2016).
- [Cas16] L. Caspani, R. P. M. Kaipurath, M. Clerici, M. Ferrera, T. Roger, J. Kim, N. Kinsey, M. Pietrzyk, A. Di Falco, V. M. Shalaev, A. Boltasseva, and D. Faccio, “Enhanced nonlinear refractive index in ϵ -near-zero materials,” *Physical Review Letters* **116**, 233901 (2016).
- [Che15] C.-H. Cheng, C.-L. Wu, Y.-H. Lin, W.-L. Yan, M.-H. Shih, J.-H. Chang, C.-I. Wu, C.-K. Leeb, and G.-R. Lin, “Strong optical nonlinearity of the nonstoichiometric silicon carbide,” *J. Mater. Chem. C* **3**, 10164 (2015).
- [Hay16] A. G. Hayrapetyan, J. B.Götte, K. K. Grigoryan, S. Fritzsche, and R. G.Petrosyan, “Electromagnetic wave propagation in spatially homogeneous yet smoothly time-varying dielectric media,” *J. of Quantitative Spectroscopy and Radiative Transfer* **178**, 158 (2016).
- [Hir79] O. Hirota and Y. Suematsu, “Noise properties of injection lasers due to reflected waves,” *IEEE Journal of Quantum Electronics* **15**, 142 (1979).
- [Hma11] H. Ma, Z. He, and K. Hotate, “Reduction of backscattering induced noise by carrier suppression in waveguide-type optical ring resonator gyro,” *J. of Lightwave Technol.* **29**, 85 (2011).
- [Jia07] S. Jiang, B. Bristiel, Y. Jaouën, P. Gallion, E. Pincemin, and S. Capouilliet, “Full characterization of modern transmission fibers for Raman amplified-based communication systems,” *Optics Express* **15**, 4883 (2007).
- [Kim16] J. Kim, A. Dutta, G. V. Naik, A. J. Giles, F. J. Bezares, C. T. Ellis, J. G. Tischler, A. M. Mahmoud, H. Caglayan, O. J. Glembocki, A. V. Kildishev, J. D. Caldwell, A. Boltasseva, and N. Engheta, “Role of epsilon-near-zero substrates in the optical response of plasmonic antennas,” *Optica* **3**, 339 (2016).
- [Leo17] F. D. Leonardis, R. A. Soref, and V. M. N. Passaro, “Dispersion of nonresonant third-order nonlinearities in silicon carbide,” *Scientific Reports* **7**, 40924 (2017).
- [Res19] O. Reshef, I. D. Leon, M. Z. Alam, and R. W. Boyd, “Nonlinear optical effects in epsilon-near-zero media,” *Nature Reviews Materials* **4**, 535 (2019).
- [Tem86] H. Temkin, N. Olsson, J. Abeles, R. Logan, and M. Panish, “Reflection noise in index-guided InGaAsP lasers,” *IEEE Journal of Quantum Electronics* **22**, 286 (1986).

2.2. I think to be suitable for publication there needs to be some concrete prospect of realising these temporal profiles experimentally.

We agree with the reviewer on the importance of this issue. Consequently, we have performed a deep literature review along this line. As a result, we have found several existing technologies that would already enable the implementation of all the proposed index variations, not only in electromagnetism, but also in acoustics and elasticity. As discussed below (points 2.3 and 2.4), the most challenging case is that of high-frequency electromagnetism, although, as mentioned in the previous point, ongoing research efforts are approaching the required technological features to realise the proposed T-SUSY modulations, some of which are already implementable with current state of the art. In the following points, we describe in detail the available technological options for realising T-SUSY modulations in each of these fields.

2.3. The idea of using non-linearity is a good one, but this will restrict Δn to small values.

While it is true that nonlinear index variations are typically small, ENZ nonlinear materials can provide a quite high refractive index excursion, since $\Delta n \approx \Delta \epsilon_r / (2n_-)$ and $n_- \rightarrow 0$ in these media [Ala16]. Moreover, as discussed in point 2.1, the parameter that determines the scattering properties of a time-varying medium is $\Delta n/n_-$, which can reach very high values at optical frequencies. For example, $\Delta n/n_- \approx 7$ for ITO [Ala16], which largely exceeds the values considered in our work ($\Delta n/n_- < 0.7$). Therefore, the implementation of the proposed modulations via strongly-nonlinear ENZ media is not actually limited by the achievable values of Δn .

Restricting ourselves to high-frequency electromagnetism (EM), the main current experimental drawback for the implementation of some of the temporal profiles reported in our work [e.g. those in Fig. 2(b) and Fig. 3(b)] would be the achievement of high $\Delta t/T_0$ ratios and, in some cases, of low optical losses [Ala16,Cas16,Res19]. Nonetheless, note that temporal index modulations with values of $\Delta n/n_-$ as high as 1 and values of $\Delta t/T_0$ as low as 5 could be attained with SiC. This is the case, for instance, of the modulations depicted in Fig. 3 of the main text.

To conclude, as mentioned above, note that strongly-nonlinear optical ENZ media with better properties could be developed in the future. Moreover, since this is a recent emerging field in which only the NIR and

MIR bands have mainly been explored [Res19], low-frequency ENZ (meta)materials providing a high (low) ratio $\Delta n/n_-$ ($\Delta t/T_0$) in combination with low linear losses could be discovered in forthcoming studies [Gli17,Zha17,Res19].

References

- [Ala16] M. Z. Alam, I. De Leon, and R. W. Boyd, “Large optical nonlinearity of indium tin oxide in its epsilon-near-zero region,” *Science* **352**, 6287 (2016).
[Cas16] L. Caspani, R. P. M. Kaipurath, M. Clerici, M. Ferrera, T. Roger, J. Kim, N. Kinsey, M. Pietrzyk, A. Di Falco, V. M. Shalaev, A. Boltasseva, and D. Faccio, “Enhanced nonlinear refractive index in ϵ -near-zero materials,” *Physical Review Letters* **116**, 233901 (2016).
[Gli17] G. Li, S. Zhang, and T. Zentgraf, “Nonlinear photonic metasurfaces,” *Nature Reviews* **2**, 17010 (2017).
[Res19] O. Reshef, I. D. Leon, M. Z. Alam, and R. W. Boyd, “Nonlinear optical effects in epsilon-near-zero media,” *Nature Reviews Materials* **4**, 535 (2019).
[Zha17] X. C. Zhang, A. Shkurinov, and Y. Zhang, “Extreme terahertz science,” *Nature Photonics* **11**, 16 (2017).

2.4. Can another higher index contrast example be given?

In high-frequency EM, an additional option to implement temporal index modulations with a high Δn is the use of multi-level phase-change materials (PCMs). As an example, germanium antimony telluride (GST) compounds can provide $\Delta n \approx 3$ with very low losses at $\lambda = 10 \mu\text{m}$ [Mic13], as well as with a controllable degree of crystallization allowing a gradual index variation between that of the amorphous and crystalline states [Wan16].

However, the minimum ratio $\Delta t/T_0$ associated with standard PCM technology is much higher than that of the strongly-nonlinear ENZ media discussed above. Specifically, the lowest typical transition time between the amorphous and crystalline states is around 400 ps, which leads to a minimum ratio of $\Delta t/T_0 \sim 12000$ [Wan08]. This is the main drawback of current PCMs concerning the implementation of the temporal profiles reported in our work. Nevertheless, it has been shown that femtosecond phase transitions based on nonthermal mechanisms can be achieved in some PCMs [Tin98]. This could be a promising route for developing faster PCM technology in the future, with which to realise the proposed T-SUSY modulations at optical frequencies.

References

- [Mic13] A.-K. U. Michel, D. N. Chigrin, T. W. W. Maß, K. Schönauer, M. Salinga, M. Wuttig, and T. Taubner, “Using low-loss phase-change materials for mid-infrared antenna resonance tuning,” *Nano Letters* **13**, 3470 (2013).
[Tin98] K. S.-Tinten, J. Solis, J. Bialkowski, J. Siegel, C. N. Afonso, and D. von der Linde, “Dynamics of ultrafast phase changes in amorphous GeSb films,” *Phys. Rev. Lett.* **81**, 3679 (1998).
[Wan08] W. J. Wang, L. P. Shia, R. Zhao, K. G. Lim, H. K. Lee, T. C. Chong, and Y. H. Wu, “Fast phase transitions induced by picosecond electrical pulses on phase change memory cells,” *Appl. Phys. Lett.* **93**, 043121 (2008).
[Wan16] Q. Wang, E. T. F. Rogers, B. Gholipour, C.-M. Wang, G. Yuan, J. Teng, and N. I. Zheludev, “Optically reconfigurable metasurfaces and photonic devices based on phase change materials,” *Nature Photonics* **10**, 60 (2016).

2.5. Are there examples of such time variations in e.g. acoustics or low frequency EM?

We thank the reviewer for her/his suggestion, which has led us to find several potential T-SUSY implementation platforms within these fields of physics, paving the way toward the realisation of different T-SUSY experiments in the short term. Let us discuss these options in detail.

Low-frequency EM

In low-frequency EM, the so-called time-varying transmission lines (TVTLs), whose effective dielectric properties vary as a function of space and time, constitute an ideal experimental platform to test and exploit all the proposed T-SUSY concepts. TVTLs usually rely on a space- and time-varying capacitance per unit length $C(z, t)$, in which case, the TVTL is governed by a wave equation of the form [Qin14]:

$$\left[\frac{\partial^2}{\partial z^2} - LC(z, t) \frac{\partial^2}{\partial t^2} \right] \Psi(z, t) = 0, \quad (\text{r2.2})$$

where L is the (constant) inductance per unit length, $\Psi(z, t) := C(z, t)v(z, t)$, and v is the voltage in the TVTL. As experimentally demonstrated in a variety of works, TVTLs with a space-time variable capacitance can be implemented by periodically loading a transmission line with a bank of distributed varactor diodes [Qin14]. The capacitance of each diode can be independently modulated in time via a control voltage. If a modulation of the form $C(z, t) = C_s(z)C_T(t)$ is used, expressing the wave function as $\Psi(z, t) = \psi(t)\phi(z)$, Eq. (r2.2) becomes:

$$\frac{1}{C_s(z)} \frac{\phi''(z)}{\phi(z)} = LC_T(t) \frac{\ddot{\psi}(t)}{\psi(t)}, \quad (\text{r2.3})$$

with $\ddot{\psi}$ being the second-order time derivative of ψ . Therefore, we must have:

$$LC_T(t) \frac{\ddot{\psi}(t)}{\psi(t)} = \gamma, \quad (\text{r2.4})$$

where γ is a constant. Assuming that $C_- := C_T(t \rightarrow -\infty)$ is also a constant, we obtain $\gamma = -\omega^2 LC_-$ for a wave with an angular frequency ω at $t \rightarrow -\infty$. This leads to a temporal Helmholtz's equation:

$$\left[\frac{d^2}{dt^2} + \omega^2 \frac{C_-}{C_T(t)} \right] \psi(t) = 0, \quad (\text{r2.5})$$

which exactly matches Eq. (3) of the R paper by defining $N^2(t) := C_-/C_T(t)$. Taking into account that Eq. (r2.2) is analogous to Eq. (1.8) of the R Supplementary Information (setting $C_S \equiv 1$), we can identify an *equivalent* time-varying refractive index $n_T(t)$ of the form:

$$\frac{n_T^2(t)}{c_0^2} \equiv LC_T(t). \quad (\text{r2.6})$$

From Eq. (r2.6), we can infer that the equivalent background index is $n_- = c_0 \sqrt{LC_-}$ and the equivalent index excursion is $\Delta n_T = c_0 \sqrt{L(C_- + \Delta C_T)} - c_0 \sqrt{LC_-}$. Hence, the normalised index excursion is:

$$\frac{\Delta n_T}{n_-} = \sqrt{1 + \frac{\Delta C_T}{C_-}} - 1. \quad (\text{r2.7})$$

Using commercial values of varactor diodes, typical excursions of $\Delta C_T \approx 12C_-$ are feasible [WebC], enabling as to attain a wide range of normalised index variations:

$$\frac{\Delta n_T}{n_-} \leq 2.5. \quad (\text{r2.8})$$

On the other hand, the ratio $\Delta t/T_0$ of this technology can be remarkably low. For instance, a modulation frequency of 4.23 GHz ($\Delta t \approx 0.24$ ns) for a signal frequency ranging from 0.45 GHz to 1.8 GHz was reported in [Qin14], yielding a minimum $\Delta t/T_0 \approx 0.1$. This means that all the proposed T-SUSY scenarios could be covered by using TVTL technology.

Lastly, as mentioned in points 2.1 and 2.3, strongly-nonlinear (meta)materials providing low losses, a high $\Delta n/n_-$ ratio, and having an ENZ band at low frequencies (thus facilitating a small $\Delta t/T_0$ ratio) could be developed in future works [Gli17,Zha17,Res19], serving as a suitable platform for the experimental implementation of rapidly- and strongly-varying temporal index modulations in low-frequency EM.

Acoustics

In acoustics, a temporal modulation of the refractive index can be induced by modifying the medium mass density (ρ) in time (see Section 7 of the R Supplementary Information). Remarkably, as stated in [Fle15]: “*The acoustic properties of materials can be modulated much more effectively than the electromagnetic ones*”.

In fact, a number of strategies to modify the acoustic mass density by different orders of magnitude can be found in the literature [Bry98,Che16,Wan15,Pop15]. For example, as reported in [Bry98], the acoustic index modulation can reach tens of percents by using magnetoacoustic crystals. More recently, it was shown that the effective mass density of an acoustic metamaterial loaded with a series of periodically distributed membranes could be dynamically tuned with an external voltage, which actively controls the properties of the membranes via electromagnets [Che16]. Notably, even near zero mass density values can be reached in this metamaterial, which would allow a large normalised index variation range similar to that of the discussed ENZ optical materials. In this case, the minimum value of $\Delta t/T_0$ cannot be inferred from the reported data.

Another option to achieve a temporal variation of the mass density is to use materials with moving parts. For instance, in [Wan15], a rotating blade was employed to build an acoustic system with a time-varying effective mass density, yielding a theoretical (experimental) ratio $\Delta n_T/n_- \approx 0.6$ ($\Delta n_T/n_- \approx 0.1$). Although a low mass density modulation frequency resulting in a value of $\Delta t/T_0 > 10$ was used in this experiment, the lower technological limit of this parameter was not reported.

Finally, there exist more sophisticated proposals in which both $\Delta n_T/n_-$ and $\Delta t/T_0$ can be widely varied and controlled by using active acoustic metamaterials [Pop15]. The local acoustic response of this kind of system can be changed almost arbitrarily in real time by configuring the digital electronics that control the

metamaterial acoustic properties. With this approach, we do not observe any limitation to implement experimentally any of the T-SUSY modulations reported in the main text.

Elasticity

Outstandingly, we have found that T-SUSY can also arise in elastic media. Specifically, consider the propagation of flexural waves in elastic beams with space- and time-varying properties, which is described by the Euler-Bernoulli equation [Ger97,Tra19]:

$$\frac{\partial}{\partial t} \left(m(\mathbf{r}, t) \frac{\partial w(\mathbf{r}, t)}{\partial t} \right) + \frac{\partial^2}{\partial x^2} \left(D(\mathbf{r}, t) \frac{\partial^2 w(\mathbf{r}, t)}{\partial x^2} \right) = 0, \quad (\text{r2.9})$$

where w is the deflection or transverse motion of the beam, and where the stiffness $D = EI$ (with E denoting Young's modulus and I the second moment of area of the beam cross section) and the beam linear mass $m = \rho A$ (with ρ the mass density and A the cross-sectional area) can be functions of space and time. For a beam with a time-independent mass (which is usually the case), assuming that the stiffness can be expressed as $D(\mathbf{r}, t) = D_S(\mathbf{r})D_T(t)$, and applying separation of variables to the deflection as $w(\mathbf{r}, t) = w_S(\mathbf{r})w_T(t)$, Eq. (r2.9) is reduced to:

$$D_T(t) \frac{\ddot{w}_T(t)}{w_T(t)} = - \frac{1}{m(\mathbf{r})w_S(\mathbf{r})} \frac{\partial^2}{\partial x^2} \left(D_S(\mathbf{r}) \frac{\partial^2 w_S(\mathbf{r})}{\partial x^2} \right). \quad (\text{r2.10})$$

Following the same reasoning as in Section 7 of the R Supplementary Information and defining:

$$n_T(t) := \frac{1}{\sqrt{D_T(t)}}, \quad (\text{r2.11})$$

with $n_- = n_T(t \rightarrow -\infty)$, we obtain a temporal Helmholtz's equation of the form:

$$\left[\frac{d^2}{dt^2} + \omega^2 \frac{n_-^2}{n_T^2(t)} \right] w_T(t) = 0, \quad (\text{r2.12})$$

which is the same equation as Eq. (3) of the R paper and Eq. (r2.5) of TVTLs.

Elastic beams with a time-varying stiffness were demonstrated in [Tra19]. The stiffness was temporally modulated with an amplitude $\alpha = \Delta D_T/D_- = 0.14$ and a modulation frequency $f_m \in [10,15]$ kHz. The frequency of the excitation signal [our wave function in Eq. (r2.12)] was 5 kHz. This corresponds to the following ratios:

$$\frac{\Delta n_T}{n_-} = \left| \frac{1}{\sqrt{1+\alpha}} - 1 \right| = 0.06; \quad (\text{r2.13})$$

$$\frac{\Delta t}{T_0} = \frac{1/f_m}{1/(5 \text{ KHz})} \in [0.3, 0.5]. \quad (\text{r2.14})$$

Note that higher values of the modulation amplitude can be achieved. For instance, an experimental value of $\alpha = 1.4$ was reported in [Air11], which corresponds to a ratio $\Delta n/n_- \approx 0.4$. These variations are of the order of the examples analysed in our manuscript, showing that the field of elasticity would also be a fertile ground for T-SUSY experiments.

Finally, we would like to point out the remarkable fact that, while T-SUSY is a natural underlying property of the Euler-Bernoulli equation, S-SUSY is not, as the spatial derivative that appears in the equation is of fourth order.

References

- [Air11] L. Airoldi and M. Ruzzene, "Design of tunable acoustic metamaterials through periodic arrays of resonant shunted piezos," *New Journal of Physics* **13**, 113010 (2011).
- [Bry98] A. P. Brysev, L. M. Krutyanski, and V. L. Preobrazhenskii, "Wave phase conjugation of ultrasonic beams," *Physics-Uspekhi* **41**, 793 (1998).
- [Che16] Z. Chen, C. Xue, L. Fan, S.-Y. Zhang, X.-J. Li, H. Zhang, and J. Ding, "A tunable acoustic metamaterial with double-negativity driven by electromagnets," *Scientific Reports* **6**, 30254 (2016).
- [Fle15] R. Fleury, D. L. Sounas, and A. Alù, "Subwavelength ultrasonic circulator based on spatiotemporal modulation," *Phys. Rev. B* **91**, 174306 (2015).
- [Ger97] J. M. Gere and S. P. Timoshenko, *Mechanics of Materials*. (PWS Publishing Company, Boston, 1997).
- [Gli17] G. Li, S. Zhang, and T. Zentgraf, "Nonlinear photonic metasurfaces," *Nature Reviews* **2**, 17010 (2017).
- [Pop15] B.-I. Popa, D. Shinde, A. Konneker, and S. A. Cummer, "Active acoustic metamaterials reconfigurable in real time," *Phys. Rev. B* **91**, 220303(R) (2015).
- [Qin14] S. Qin, Q. Xu, and Y. E. Wang, "Nonreciprocal components with distributedly modulated capacitors," *IEEE Transactions on Microwave Theory and Techniques* **62**, 2260 (2014).

- [Res19] O. Reshef, I. D. Leon, M. Z. Alam, and R. W. Boyd, “Nonlinear optical effects in epsilon-near-zero media,” *Nature Reviews Materials* **4**, 535 (2019).
- [Tra19] G. Trainiti, Y. Xia, J. Marconi, G. Cazzulani, A. Erturk, and M. Ruzzene, “Time-periodic stiffness modulation in elastic metamaterials for selective wave filtering: theory and experiment,” *Phys. Rev. Lett.* **122**, 124301 (2019).
- [Wan15] Q. Wang, Y. Yang, X. Ni, Y.-L. Xu, X.-C. Sun, Z.-G. Chen, L. Feng, X.-P. Liu, M.-Hui Lu, and Y.-F. Chen, “Acoustic asymmetric transmission based on time-dependent dynamical scattering,” *Scientific Reports* **5**, 10880 (2015).
- [WebC] Findchips, <https://www.findchips.com/parametric/Diodes/Varactors> (2019).
- [Zha17] X. C. Zhang, A. Shkurinov, and Y. Zhang, “Extreme terahertz science,” *Nature Photonics* **11**, 16 (2017).

2.6. Modifications in R paper & R supplementary

Following the referee’s suggestions, we have extended the “Discussion” Section in the revised manuscript, which now includes a detailed analysis on the different possibilities for realising experimental T-SUSY modulations in high-frequency EM, low-frequency EM, acoustics, and elasticity (see pages 12 and 13). To facilitate the rapid assessment of the suitability of each experimental platform for the implementation of a certain T-SUSY index, we have also included a table summarising the main characteristics of all the analysed technologies in an easy to read manner (see Table 1 “Potential T-SUSY implementation platforms” on page 14 of the R paper). For completeness, we also provide all the required theory and discussions about TVTLs, acoustics, and elasticity in Sections 6, 7 and 8 of the R Supplementary Information, respectively.

Furthermore, bearing in mind that the scattering properties of the system only depend on $n(t)/n_-$, we have expressed the analysed index profiles in terms of this quantity in the revised version of the manuscript, so that they are independent of the background index. This includes the update of Figs. 2 and 3 of the R paper, Supplementary Figs. 3.3, 3.7 and 3.11, and Supplementary Movies 1-4.

2.7. Final remarks

After addressing how the proposed T-SUSY index modulations could be implemented with current experimental platforms, we would like to conclude our answer to this question by highlighting the relevance of our results in relation to future photonics research challenges.

In the years to come, the necessity of reducing the size of basic signal processing devices, such as phase shifters and optical modulators, will be intimately related to light manipulation via rapidly- and strongly-varying temporal modulations of the refractive index, as suggested in [Res19,Woo18,Fei10,Eli18]. As the modulation speed and amplitude increase, the reflection generated by temporal scattering could become one of the main physical impairments associated with light propagation in future photonic systems. Hence, it will be crucial to be able to engineer the profile of the employed modulations so as to minimize this undesired effect.

In such a scenario, T-SUSY could emerge as a powerful design tool, as it provides a simple analytical solution to find an *infinite* set of all-dielectric, isotropic, omnidirectional, polarisation-independent, and non-complex *reflectionless* temporal index modulations. As mentioned in the manuscript, no known previous medium possesses all these attributes, which is a remarkable feature, considering that the quest for non-trivial reflectionless optical media has traditionally been a topic of extreme importance in photonics.

Moreover, it is worth emphasizing the high versatility of T-SUSY, conferred by the possibility of modifying $\Delta n/n_-$ and $\Delta t/T_0$ without altering the intensity-scattering behaviour of the supersymmetric modulations (see e.g. Supplementary Figs. 3.1 and 3.2), and by the fact that n_- is a degree of freedom in this theory. As a result, we can easily calculate reflectionless time-varying index profiles accommodated to each specific technology.

Finally, we would like to stress the interest of the T-SUSY formalism for systems with discrete optical spectra. As shown in the manuscript, this theory would enable the creation of devices with relevant applications, such as the versatile and reconfigurable pulse-shape transformers addressed in our work (also highly important photonic components [Wei11]). In this case, only extremely low index perturbations ($\Delta n \sim 10^{-12}$) are required to implement the proposed T-SUSY modulations, avoiding the associated technological challenges discussed above.

References

- [Eli18] E. Li, Q. Gao, R. T. Chen, and A. X. Wang, “Ultracompact silicon-conductive oxide nanocavity modulator with 0.02 lambda-cubic active volume,” *Nano Letters* **18**, 1075 (2018).
- [Fei10] E. Feigenbaum, K. Diest, and H. A. Atwater, “Unity-order index change in transparent conducting oxides at visible frequencies,” *Nano Letters* **10**, 2111 (2010).
- [Res19] O. Reshef, I. D. Leon, M. Z. Alam, and R. W. Boyd, “Nonlinear optical effects in epsilon-near-zero media,” *Nature Reviews Materials* **4**, 535 (2019).

[Wei11] A. Weiner, “Ultrafast optical pulse shaping: a tutorial review,” *Optics Communications* **284**, 3669 (2011).
 [Woo18] M. G. Wood, S. Campione, S. Parameswaran, T. S. Luk, J. R. Wendt, D. K. Serkland, and G. A. Keeler, “Gigahertz speed operation of epsilon-near-zero silicon photonic modulators,” *Optica* **5**, 233 (2018).

3. The restriction to a fixed initial frequency ω_0 seems quite severe. I think it is important to spell out in more detail what this restriction implies. For example, it is also stated that this can be the “central frequency” in the non-monochromatic regime, but it was not clear to me whether this was an approximation, and if so under what conditions it holds.

After critically reviewing the manuscript in relation to the referee's comment, we agree that this is a subtle point that required a more detailed explanation.

Indeed, the prescription for the T-SUSY superpartner n_{T2} of a medium having a temporal index n_{T1} depends on the frequency of the electromagnetic field ω [as inferred, e.g., from Eq. (4) of the main text]. Since it would be challenging to realise such a frequency-dependent index time variation in practice, we consider a more realistic and typical scenario in which the medium has the same variation for all relevant frequencies of the electromagnetic field. This variation is taken to be the one that makes n_{T1} and n_{T2} supersymmetric at a fixed frequency ω_0 , which will be a free design parameter (it can be, e.g., the central frequency of the spectral band of interest, or any other reference frequency providing the desired response). That is, we suppress the frequency dependence of n_{T2} by fixing ω to ω_0 in Eq. (4). Consequently, the two systems will be guaranteed to be exact T-SUSY partners at ω_0 , while they may exhibit different properties at other frequencies. The same situation is found in S-SUSY [Mir13,Mac18].

Nevertheless, it can be shown that the T-SUSY connection holds almost exactly in a broad frequency band, which is a typical feature of supersymmetric optical systems [Mir14,Mac18]. To verify this, one needs to calculate the response associated with the index n_{T2} for $\omega \neq \omega_0$. In the case of scattering problems, this response will be characterised by the frequency-dependent complex reflection and transmission coefficients of the system, which can be rigorously calculated by solving numerically the wave equation associated with n_{T2} . The T-SUSY partner of a constant refractive index analysed in the manuscript, whose scattering coefficients depicted in Fig. 2(e) were obtained by solving numerically Supplementary Eq. (1.8), is an illustrative example of these ideas. Since the constant index n_{T1} is reflectionless, n_{T2} is also expected to be reflectionless at a frequency $\omega = \omega_0$, for which n_{T1} and n_{T2} are exact T-SUSY partners. Figure 2(e) not only confirms this, but also demonstrates that the reflectionless property is almost preserved in a wide spectral band. For example, for $\Omega = 2\omega_0^2$ (which corresponds to a rapidly-varying index with $\Delta t/T_0 = 0.6$), the system has a reflectance $|R_2|^2 < 10^{-3}$ (−30 dB) in a fractional bandwidth $F = \Delta\omega/\omega_0 = 30\%$.

Moreover, as mentioned in the manuscript, it is possible to tune the value of F (as well as the phase shift introduced by the system) via the free design parameter Ω , which allows us to engineer n_{T2} so as to satisfy the particular specifications of different applications. Concretely, F decreases as Ω increases. For instance, $|R_2|^2 < 10^{-3}$ in a fractional bandwidth $F > 55\%$ for $\Omega = 1.5\omega_0^2$, while $|R_2|^2 < 10^{-3}$ in a fractional bandwidth $F \approx 8\%$ for $\Omega = 6\omega_0^2$. Consequently, having a fixed initial design frequency ω_0 will not pose a practical limitation in most situations.

We have clarified all these points in the new version of the manuscript via the following changes:

- We have modified the main text after Eq. (3), which now reads: “where $N^2(t) := n_-^2/n_T^2(t)$, $n_- := n_T(t \rightarrow -\infty)$, and ω is the angular frequency of the field at $t \rightarrow -\infty$. For a polychromatic wave, the total field is given by the superposition of the solutions to equation (3) for each spectral component (value of ω).”
- On page 5 of the R paper, we have included the following explanation: “As seen, the prescription for $n_{T2}(t)$ depends on ω . Since it would be challenging to realise such a frequency-dependent index in practice, we consider a more realistic and typical scenario in which the medium has the same temporal variation for all relevant frequencies of the electromagnetic field. This variation is taken to be the one that makes $n_{T1}(t)$ and $n_{T2}(t)$ supersymmetric at a frequency ω_0 , which will be a free design parameter (it can be, e.g., the central frequency of the spectral band of interest, or any other reference frequency providing the desired response). That is, we suppress the frequency dependence by fixing ω to ω_0 in equation (4). Consequently, the two systems will be guaranteed to be exact T-SUSY partners at ω_0 , while they may exhibit different properties at other frequencies. The same situation is found in S-SUSY^{9,12}. Nevertheless, it can be shown that the T-SUSY connection generally holds almost exactly in a broad frequency band, which is a typical feature of supersymmetric optical systems^{10,12}. This can be verified by calculating the solution to the wave equation associated with n_{T2} at $\omega \neq \omega_0$ (Supplementary Section 1), from which the medium spectral response can be obtained (an example is given below for a reflectionless system, see Fig. 2).”

- Finally, we have added the following discussion on page 7 of the R paper: “As discussed above, another general feature of T-SUSY is that it is only exact for the design frequency $\omega = \omega_0$. Therefore, n_2 will be invisible ($R_2 = 0$, $|T_2| = 1$) for all directions and polarisations at ω_0 , while a reflected wave will appear at other frequencies. The response of n_2 at $\omega \neq \omega_0$ will be characterised by the value of R_2 and T_2 as a function of frequency, which can be rigorously obtained by solving numerically Supplementary equation (1.8). The result for the present example is depicted in Fig. 2(e), which not only confirms the expected reflectionlessness of n_2 at $\omega = \omega_0$, but also demonstrates that this property is almost preserved in a wide spectral band. For example, for $\Omega = 2\omega_0^2$ (which corresponds to a rapidly-varying index with $\Delta t/T_0 = 0.6$, $T_0 = 2\pi/\omega_0$), the system has a reflectance $|R_2|^2 < 10^{-3}$ in a fractional bandwidth $F = \Delta\omega/\omega_0 = 30\%$. Moreover, the spectral span for which n_2 is almost invisible ($R_2 \approx 0$, $|T_2| \approx 1$) can also be tailored via Ω , enabling us to generate custom-made transparent temporal windows within n_2 only for desired bands [Figs. 1(a) and 2]. Concretely, F decreases as Ω increases. For instance, $|R_2|^2 < 10^{-3}$ in a fractional bandwidth $F > 55\%$ for $\Omega = 1.5\omega_0^2$, while $|R_2|^2 < 10^{-3}$ in a bandwidth $F = 8\%$ for $\Omega = 6\omega_0^2$. Notably, out of the invisible band, all waves are (partially) retroreflected along the input path, in contrast to spatial retroreflectors, in which the reflected path is parallel to, but different from, the input one³⁰.”

References

- [Mir13] M.-A. Miri, M. Heinrich, R. El-Ganainy, and D. N. Christodoulides, “Supersymmetric optical structures,” *Phys. Rev. Lett.* **110**, 233902 (2013).
- [Mir14] M.-A. Miri, M. Heinrich, and D. N. Christodoulides, “SUSY-inspired one-dimensional transformation optics,” *Optica* **1**, 89 (2014).
- [Mac18] A. Macho, R. Llorente and C. García-Meca, “Supersymmetric transformations in optical fibers,” *Phys. Rev. Applied* **9**, 014024 (2018).

Feedback from Reviewer #3 Comments

General comments:

The manuscript “Optical Supersymmetry in the Time Domain” demonstrates time domain version (T-SUSY) of optical SUSY. The authors numerically and analytically show a new paradigm of optical SUSY in time domain with potential applications such as temporally controlled e.g. phase shifter and a frequency converter. The paper is well-written and the results and discussion are sound and convincing. However, I would like the two notes below are addressed before publication.

The authors sincerely thank the reviewer for highlighting the potential of our results and for his/her constructive comments and suggestions.

1. Can the authors exemplify a material with a temporal change in refractive index as shown in Fig. 2b inset ($\Delta n = 2-1.2 = 0.8$)? If not could they replace Fig. 2 with real material values?

In high-frequency electromagnetism (EM), we know of two material families that can exhibit a temporal change in refractive index of this order:

1. Strongly-nonlinear epsilon-near zero (ENZ) materials: the most relevant example of this kind is indium tin oxide (ITO), which can undergo refractive index variations as high as $\Delta n = 0.72$ in its ENZ band (at free-space wavelengths $\lambda_{\text{ENZ}} \approx 1240$ nm).
2. Phase-change materials: a paradigmatic example is provided by germanium antimony telluride (GST) compounds, which have a controllable degree of crystallization that enables a gradual index variation between that of its amorphous and crystalline states, with a maximum $\Delta n \approx 3$ [Wan16]. Additionally, GST possesses very low losses at $\lambda \approx 10 \mu\text{m}$ [Mic13].

Nonetheless, while the refractive index variations of the proposed T-SUSY modulations could be achieved with these materials, there is another aspect that has to be taken into account, namely, the medium response time Δt (defined as the required time interval to obtain the maximum index variation Δn). Certainly, the stronger effects arise when Δt is of the order of the period of the optical signal (T_0) or lower. However, in the case of ITO, $\Delta t \approx 36T_0$ ($\Delta t \in [150,200]$ fs, $T_0 \approx 4$ fs) [Ala16], while for GST, $\Delta t/T_0 \sim 12000$ ($T_0 \approx 33$ fs, and typically $\Delta t \geq 400$ ps) [Wan08]. Most of the T-SUSY modulations reported in our work could not be implemented with these materials due to their insufficiently fast response times, including the one depicted in Fig. 2(b), for which $\Delta t/T_0 = 0.6$. Additionally, ITO presents non-negligible losses for very low index values.

On the other hand, it is important to point out that the temporal response of a system depends on the variation of n/n_- , rather than on that of n alone [Hay16]. This can be seen by noting that the only medium parameter entering the temporal Helmholtz’s equation is $N = n_-/n$. Similarly, defining the normalised time as $t_N := t/T_0$, this equation can be recast as:

$$\left[\frac{d^2}{dt_N^2} + (2\pi)^2 \frac{n_-^2}{n_T^2(T_0 t_N)} \right] \psi(T_0 t_N) = 0, \quad (\text{r3.1})$$

That is, the evolution of ψ expressed as a function of t_N (and hence the associated temporal scattering coefficients) will be equal for any system having the same normalised index variation $n_T(T_0 t_N)/n_-$. This implies that there exist infinite modulations with an equivalent response, and that the set of functionalities achievable by a certain technology will be determined by the maximum (minimum) allowed values of $\Delta n/n_-$ ($\Delta t/T_0$). For instance, any material supporting index changes with $\Delta n/n_- \approx 0.4$ and $\Delta t/T_0 \approx 0.6$ can be used to implement the same functionality as the modulation of Fig. 2(b).

Bearing in mind the comments of the reviewers with respect to the experimental implementation of the developed theory, and as we completely agree on the importance of this issue, we have assessed the suitability of a number of technologies for the practical realisation of the index variations required by the proposed T-SUSY modulations (including that of Fig. 2b) in terms of $\Delta n/n$ and $\Delta t/T_0$. In the following, we describe in detail the available technological options in the fields of high-frequency EM, low-frequency EM, acoustics, and elasticity. Subsequently, we discuss the modifications made to the R version of the paper in accordance with the conclusions of our analysis.

High-frequency EM

As mentioned above, the most relevant platforms we have analysed in this case are:

1. Strongly-nonlinear ENZ media: in the near-infrared (NIR), materials such as ITO and aluminium-doped zinc oxide (AZO) provide large values of $\Delta n/n$ (7.2 and 4.5, respectively), but are limited by their high values of $\Delta t/T_0$ (36 and 33, respectively) and their linear losses [Ala16,Cas16]. As a potentially better alternative, we have also explored nonlinear ENZ materials in the mid-infrared (MIR). One of the most interesting members of this group is silicon carbide (SiC), with an estimated response time $\Delta t/T_0 \approx 5$ [Kim16,Che15] and an estimated maximum normalised index variation of $\Delta n/n_- \approx 1$ in its ENZ band ($\lambda_{\text{ENZ}} = 10.33 \mu\text{m}$) [Che15,Leo17,Cas16].³ This implies that modulations of the kind of those depicted in Fig. 3 of the manuscript could be potentially realised with SiC, whilst the modulation of Fig. 2(b) would also be out of reach for this material. On the other hand, it is worth mentioning that, as suggested in [Res19]:

“There might, of course, be a better material platform that is still to be discovered. The ideal material would have the following properties: CMOS compatibility, high degree of crystallinity, large carrier mobility and low linear losses. If possible, this material would have also a tailorable zero-permittivity wavelength, and it would be straightforward to deposit and nanostructure.”

Therefore, strongly-nonlinear optical ENZ media with better properties could be developed in the future. Moreover, since this is a recent emerging field in which only the NIR and MIR bands have mainly been explored [Res19], low-frequency ENZ (meta)materials providing a high (low) ratio $\Delta n/n_-$ ($\Delta t/T_0$) in combination with low linear losses could be discovered in forthcoming studies [Gli17,Zha17,Res19].

2. Phase change materials: multi-level PCMs also emerge as a potential platform to obtain large values of $\Delta n/n_-$ in high-frequency EM. As mentioned above, GST has a $\Delta n \approx 3$ ($\Delta n/n_- \approx 0.9$) with low losses at $\lambda = 10 \mu\text{m}$ [Mic13] and a gradual index variation [Wan16]. However, the large response time of PCMs in the NIR and MIR ($\Delta t \geq 400$ ps [Wan08], $\Delta t/T_0 \gg 1$) is currently the main drawback of this technology. Nevertheless, it has been shown that femtosecond phase transitions based on nonthermal mechanisms can be achieved in some PCMs [Tin98]. This could be a promising route for developing faster PCM technology in the future, with which to realise the proposed T-SUSY modulations at optical frequencies.

Low-frequency EM

In low-frequency EM, the so-called time-varying transmission lines (TVTLs), whose effective dielectric properties vary as a function of space and time, constitute an ideal experimental platform to test and exploit all the proposed T-SUSY concepts. TVTLs usually rely on a space- and time-varying capacitance per unit length $C(z, t)$, in which case, the TVTL is governed by a wave equation of the form [Qin14]:

$$\left[\frac{\partial^2}{\partial z^2} - LC(z, t) \frac{\partial^2}{\partial t^2} \right] \Psi(z, t) = 0, \quad (\text{r3.2})$$

where L is the (constant) inductance per unit length, $\Psi(z, t) := C(z, t)v(z, t)$, and v is the voltage in the TVTL. As experimentally demonstrated in a variety of works, TVTLs with a space-time variable capacitance can be implemented by periodically loading a transmission line with a bank of distributed varactor diodes [Qin14]. The capacitance of each diode can be independently modulated in time via a control voltage. If a modulation of the form $C(z, t) = C_S(z)C_T(t)$ is used, expressing the wave function as $\Psi(z, t) = \psi(t)\phi(z)$, Eq. (r3.2) becomes:

$$\frac{1}{C_S(z)} \frac{\phi''(z)}{\phi(z)} = LC_T(t) \frac{\ddot{\psi}(t)}{\psi(t)}, \quad (\text{r3.3})$$

with $\ddot{\psi}$ being the second-order time derivative of ψ . Therefore, we must have:

$$LC_T(t) \frac{\ddot{\psi}(t)}{\psi(t)} = \gamma, \quad (\text{r3.4})$$

where γ is a constant. Assuming that $C_- := C_T(t \rightarrow -\infty)$ is also a constant, we obtain $\gamma = -\omega^2 LC_-$ for a wave with an angular frequency ω at $t \rightarrow -\infty$. This leads to a temporal Helmholtz's equation:

³ We have estimated $\Delta t/T_0$ in SiC by considering a similar rise time to that of the ITO and AZO. In addition, $\Delta n/n_-$ has been estimated in SiC by using the general theory reported in [Cas16] along with the SiC properties reported in [Che15,Leo17].

$$\left[\frac{d^2}{dt^2} + \omega^2 \frac{C_-}{C_T(t)} \right] \psi(t) = 0, \quad (\text{r3.5})$$

which exactly matches Eq. (3) of the R paper by defining $N^2(t) := C_-/C_T(t)$. Taking into account that Eq. (r3.2) is analogous to Eq. (1.8) of the R Supplementary Information (setting $C_S \equiv 1$), we can identify an *equivalent* time-varying refractive index $n_T(t)$ of the form:

$$\frac{n_T^2(t)}{c_0^2} \equiv LC_T(t), \quad (\text{r3.6})$$

From Eq. (r3.6), we can infer that the equivalent background index is $n_- = c_0\sqrt{LC_-}$ and the equivalent index excursion is $\Delta n_T = c_0\sqrt{L(C_- + \Delta C_T)} - c_0\sqrt{LC_-}$. Hence, the normalised index excursion is:

$$\frac{\Delta n_T}{n_-} = \sqrt{1 + \frac{\Delta C_T}{C_-}} - 1. \quad (\text{r3.7})$$

Using commercial values of varactor diodes, typical excursions of $\Delta C_T \approx 12C_-$ are feasible [WebC], enabling as to attain a wide range of normalised index variations:

$$\frac{\Delta n_T}{n_-} \leq 2.5. \quad (\text{r3.8})$$

On the other hand, the ratio $\Delta t/T_0$ of this technology can be remarkably low. For instance, a modulation frequency of 4.23 GHz ($\Delta t \approx 0.24$ ns) for a signal frequency ranging from 0.45 GHz to 1.8 GHz was reported in [Qin14], yielding a minimum $\Delta t/T_0 \approx 0.1$. This means that all the proposed T-SUSY scenarios could be covered by using TVTL technology.

Lastly, as commented above, strongly-nonlinear (meta)materials providing low losses, a high $\Delta n/n_-$ ratio, and having an ENZ band at low frequencies (thus facilitating a small $\Delta t/T_0$ ratio) could be developed in future works [Gli17,Zha17,Res19], serving as a suitable platform for the experimental implementation of rapidly- and strongly-varying temporal index modulations in low-frequency EM.

Acoustics

In acoustics, a temporal modulation of the refractive index can be induced by modifying the medium mass density (ρ) in time (see Section 7 of the R Supplementary Information). Remarkably, as stated in [Fle15]: “*The acoustic properties of materials can be modulated much more effectively than the electromagnetic ones*”.

In fact, a number of strategies to modify the acoustic mass density by different orders of magnitude can be found in the literature [Bry98,Che16,Wan15,Pop15]. For example, as reported in [Bry98], the acoustic index modulation can reach tens of percents by using magnetoacoustic crystals. More recently, it was shown that the effective mass density of an acoustic metamaterial loaded with a series of periodically distributed membranes could be dynamically tuned with an external voltage, which actively controls the properties of the membranes via electromagnets [Che16]. Notably, even near zero mass density values can be reached in this metamaterial, which would allow a large normalised index variation range similar to that of the discussed ENZ optical materials. In this case, the minimum value of $\Delta t/T_0$ cannot be inferred from the reported data.

Another option to achieve a temporal variation of the mass density is to use materials with moving parts. For instance, in [Wan15], a rotating blade was employed to build an acoustic system with a time-varying effective mass density, yielding a theoretical (experimental) ratio $\Delta n_T/n_- \approx 0.6$ ($\Delta n_T/n_- \approx 0.1$). Although a low mass density modulation frequency resulting in a value of $\Delta t/T_0 > 10$ was used in this experiment, the lower technological limit of this parameter was not reported.

Finally, there exist more sophisticated proposals in which both $\Delta n_T/n_-$ and $\Delta t/T_0$ can be widely varied and controlled by using active acoustic metamaterials [Pop15]. The local acoustic response of this kind of system can be changed almost arbitrarily in real time by configuring the digital electronics that control the metamaterial acoustic properties. With this approach, we do not observe any limitation to implement experimentally any of the T-SUSY modulations reported in the main text.

Elasticity

Outstandingly, we have found that T-SUSY can also arise in elastic media. Specifically, consider the propagation of flexural waves in elastic beams with space- and time-varying properties, which is described by the Euler-Bernoulli equation [Ger97,Tra19]:

$$\frac{\partial}{\partial t} \left(m(\mathbf{r}, t) \frac{\partial w(\mathbf{r}, t)}{\partial t} \right) + \frac{\partial^2}{\partial x^2} \left(D(\mathbf{r}, t) \frac{\partial^2 w(\mathbf{r}, t)}{\partial x^2} \right) = 0, \quad (\text{r3.9})$$

where w is the deflection or transverse motion of the beam, and where the stiffness $D = EI$ (with E denoting Young's modulus and I the second moment of area of the beam cross section) and the beam linear mass $m = \rho A$ (with ρ the mass density and A the cross-sectional area) can be functions of space and time. For a beam with a time-independent mass (which is usually the case), assuming that the stiffness can be expressed as $D(\mathbf{r}, t) = D_S(\mathbf{r})D_T(t)$, and applying separation of variables to the deflection as $w(\mathbf{r}, t) = w_S(\mathbf{r})w_T(t)$, Eq. (r3.9) is reduced to:

$$D_T(t) \frac{\ddot{w}_T(t)}{w_T(t)} = - \frac{1}{m(\mathbf{r})w_S(\mathbf{r})} \frac{\partial^2}{\partial x^2} \left(D_S(\mathbf{r}) \frac{\partial^2 w_S(\mathbf{r})}{\partial x^2} \right). \quad (\text{r3.10})$$

Following the same reasoning as in Section 7 of the R Supplementary Information and defining:

$$n_T(t) := \frac{1}{\sqrt{D_T(t)}}, \quad (\text{r3.11})$$

with $n_- = n_T(t \rightarrow -\infty)$, we obtain a temporal Helmholtz's equation of the form:

$$\left[\frac{d^2}{dt^2} + \omega^2 \frac{n_-^2}{n_T^2(t)} \right] w_T(t) = 0, \quad (\text{r3.12})$$

which is the same equation as Eq. (3) of the R paper and Eq. (r3.5) of TVTLs.

Elastic beams with a time-varying stiffness were demonstrated in [Tra19]. The stiffness was temporally modulated with an amplitude $\alpha = \Delta D_T/D_- = 0.14$ and a modulation frequency $f_m \in [10,15]$ kHz. The frequency of the excitation signal [our wave function in Eq. (r3.12)] was 5 kHz. This corresponds to the following ratios:

$$\frac{\Delta n_T}{n_-} = \left| \frac{1}{\sqrt{1+\alpha}} - 1 \right| = 0.06; \quad (\text{r3.13})$$

$$\frac{\Delta t}{T_0} = \frac{1/f_m}{1/(5 \text{ KHz})} \in [0.3,0.5]. \quad (\text{r3.14})$$

Note that higher values of the modulation amplitude can be achieved. For instance, an experimental value of $\alpha = 1.4$ was reported in [Air11], which corresponds to a ratio $\Delta n/n_- \approx 0.4$. These variations are of the order of the examples analysed in our manuscript, showing that the field of elasticity would also be a fertile ground for T-SUSY experiments.

Finally, we would like to point out the remarkable fact that, while T-SUSY is a natural underlying property of the Euler-Bernoulli equation, S-SUSY is not, as the spatial derivative that appears in the equation is of fourth order.

Modifications in R paper & R supplementary

According to our previous discussion, we have expressed the analysed index profiles in terms of $n(t)/n_-$ and t/T_0 in the revised version of the manuscript, so that they are now independent of the operation frequency (inverse of T_0) and background index. This includes the update of Figs. 2 and 3, Supplementary Figs. 3.3, 3.7 and 3.11, and Supplementary Movies 1-4. With this modification, one can readily assess the capability of any technology for the implementation of the proposed modulations.

Focusing on the (normalised) temporal change in refractive index of Fig. 2(b) mentioned by the reviewer, we conclude from our analysis that it can be realised with electromagnetic TVTL technology, acoustic metamaterials, and elastic beams. Therefore, we have retained the normalised index excursion $\Delta n/n_- = 0.4$ and response time $\Delta t/T_0 = 0.6$ of the previous version of this figure, indicating the aforementioned implementation options offered by current technology in the revised manuscript (see below). At optical frequencies, although the index excursion associated with this modulation can be achieved via strongly-nonlinear ENZ media and PCMs, the required response time is still beyond the limits of these technologies. Nonetheless, other modulations such as those in Fig. 3 could be implemented at high

electromagnetic frequencies with nonlinear materials like SiC. Moreover, given the recent and promising advances in temporal index modulation technology [Res19,Fei10,Woo18,Eli18], we could expect the ratio $\Delta t/T_0$ to approach the value of Fig. 2(b) at optical frequencies with the development of new materials.

Finally, we have extended the “Discussion” Section in order to include the main points of our analysis on the different possibilities for realising experimental T-SUSY modulations in high-frequency EM, low-frequency EM, acoustics, and elasticity (see pages 12 and 13). To facilitate the rapid assessment of the suitability of each experimental platform for the implementation of a certain T-SUSY index, we have also added a table summarising the main characteristics of all the analysed technologies in an easy to read manner (see Table 1 “Potential T-SUSY implementation platforms” on page 14 of the R paper). For completeness, we provide all the required theory and discussions about TVTLs, acoustics, and elasticity in Sections 6, 7 and 8 of the R Supplementary Information, respectively.

References

- [Air11] L. Airoldi and M. Ruzzene, “Design of tunable acoustic metamaterials through periodic arrays of resonant shunted piezos,” *New Journal of Physics* **13**, 113010 (2011).
- [Ala16] M. Z. Alam, I. De Leon, and R. W. Boyd, “Large optical nonlinearity of indium tin oxide in its epsilon-near-zero region,” *Science* **352**, 6287 (2016).
- [Bry98] A. P. Brysev, L. M. Krutyanski, and V. L. Preobrazhenskii, “Wave phase conjugation of ultrasonic beams,” *Physics-Uspekhi* **41**, 793 (1998).
- [Cas16] L. Caspani, R. P. M. Kaipurath, M. Clerici, M. Ferrera, T. Roger, J. Kim, N. Kinsey, M. Pietrzyk, A. Di Falco, V. M. Shalaev, A. Boltasseva, and D. Faccio, “Enhanced nonlinear refractive index in ϵ -near-zero materials,” *Physical Review Letters* **116**, 233901 (2016).
- [Che15] C.-H. Cheng, C.-L. Wu, Y.-H. Lin, W.-L. Yan, M.-H. Shih, J.-H. Chang, C.-I. Wu, C.-K. Leeb, and G.-R. Lin, “Strong optical nonlinearity of the nonstoichiometric silicon carbide,” *J. Mater. Chem. C* **3**, 10164 (2015).
- [Che16] Z. Chen, C. Xue, L. Fan, S.-Y. Zhang, X.-J. Li, H. Zhang, and J. Ding, “A tunable acoustic metamaterial with double-negativity driven by electromagnets,” *Scientific Reports* **6**, 30254 (2016).
- [Eli18] E. Li, Q. Gao, R. T. Chen, and A. X. Wang, “Ultracompact silicon-conductive oxide nanocavity modulator with 0.02 lambda-cubic active volume,” *Nano Letters* **18**, 1075 (2018).
- [Fei10] E. Feigenbaum, K. Diest, and H. A. Atwater, “Unity-order index change in transparent conducting oxides at visible frequencies,” *Nano Letters* **10**, 2111 (2010).
- [Fle15] R. Fleury, D. L. Sounas, and A. Alù, “Subwavelength ultrasonic circulator based on spatiotemporal modulation,” *Phys. Rev. B* **91**, 174306 (2015).
- [Ger97] J. M. Gere and S. P. Timoshenko, *Mechanics of Materials*. (PWS Publishing Company, Boston, 1997).
- [Gli17] G. Li, S. Zhang, and T. Zentgraf, “Nonlinear photonic metasurfaces,” *Nature Reviews* **2**, 17010 (2017).
- [Hay16] A. G. Hayrapetyan, J. B.Götte, K. K. Grigoryan, S. Fritzsche, and R. G. Petrosyan, “Electromagnetic wave propagation in spatially homogeneous yet smoothly time-varying dielectric media,” *J. of Quantitative Spectroscopy and Radiative Transfer* **178**, 158 (2016).
- [Kim16] J. Kim, A. Dutta, G. V. Naik, A. J. Giles, F. J. Bezares, C. T. Ellis, J. G. Tischler, A. M. Mahmoud, H. Caglayan, O. J. Glembocki, A. V. Kildishev, J. D. Caldwell, A. Boltasseva, and N. Engheta, “Role of epsilon-near-zero substrates in the optical response of plasmonic antennas,” *Optica* **3**, 339 (2016).
- [Leo17] F. D. Leonardis, R. A. Soref, and V. M. N. Passaro, “Dispersion of nonresonant third-order nonlinearities in silicon carbide,” *Scientific Reports* **7**, 40924 (2017).
- [Mic13] A.-K. U. Michel, D. N. Chigrin, T. W. W. Maß, K. Schönauer, M. Salinga, M. Wuttig, and T. Taubner, “Using low-loss phase-change materials for mid-infrared antenna resonance tuning,” *Nano Letters* **13**, 3470 (2013).
- [Pop15] B.-I. Popa, D. Shinde, A. Konneker, and S. A. Cummer, “Active acoustic metamaterials reconfigurable in real time,” *Phys. Rev. B* **91**, 220303(R) (2015).
- [Qin14] S. Qin, Q. Xu, and Y. E. Wang, “Nonreciprocal components with distributedly modulated capacitors,” *IEEE Transactions on Microwave Theory and Techniques* **62**, 2260 (2014).
- [Res19] O. Reshef, I. D. Leon, M. Z. Alam, and R. W. Boyd, “Nonlinear optical effects in epsilon-near-zero media,” *Nature Reviews Materials* **4**, 535 (2019).
- [Tin98] K. S.-Tinten, J. Solis, J. Bialkowski, J. Siegel, C. N. Afonso, and D. von der Linde, “Dynamics of ultrafast phase changes in amorphous GeSb films,” *Phys. Rev. Lett.* **81**, 3679 (1998).
- [Tra19] G. Trainiti, Y. Xia, J. Marconi, G. Cazzulani, A. Erturk, and M. Ruzzene, “Time-periodic stiffness modulation in elastic metamaterials for selective wave filtering: theory and experiment,” *Phys. Rev. Lett.* **122**, 124301 (2019).
- [Wan08] W. J. Wang, L. P. Shia, R. Zhao, K. G. Lim, H. K. Lee, T. C. Chong, and Y. H. Wu, “Fast phase transitions induced by picosecond electrical pulses on phase change memory cells,” *Appl. Phys. Lett.* **93**, 043121 (2008).
- [Wan15] Q. Wang, Y. Yang, X. Ni, Y.-L. Xu, X.-C. Sun, Z.-G. Chen, L. Feng, X.-P. Liu, M.-Hui Lu, and Y.-F. Chen, “Acoustic asymmetric transmission based on time-dependent dynamical scattering,” *Scientific Reports* **5**, 10880 (2015).
- [Wan16] Q. Wang, E. T. F. Rogers, B. Gholipour, C.-M. Wang, G. Yuan, J. Teng, and N. I. Zheludev, “Optically reconfigurable metasurfaces and photonic devices based on phase change materials,” *Nature Photonics* **10**, 60 (2016).
- [WebC] Findchips, <https://www.findchips.com/parametric/Diodes/Varactors> (2019).
- [Woo18] M. G. Wood, S. Campione, S. Parameswaran, T. S. Luk, J. R. Wendt, D. K. Serkland, and G. A. Keeler, “Gigahertz speed operation of epsilon-near-zero silicon photonic modulators,” *Optica* **5**, 233 (2018).
- [Zha17] X. C. Zhang, A. Shkurinov, and Y. Zhang, “Extreme terahertz science,” *Nature Photonics* **11**, 16 (2017).

2. Authors show that T-SUSY equations are uncoupled from space. Accordingly, it is claimed that T-SUSY is polarization-independent and omnidirectional (page 4) which is unlike one-dimensional S-SUSY (off course this is true, since there is no direction limitation for time). As an example in page 6, a constant refractive index material with no reflection is considered. I assume that the material is infinite that has no boundary with free space. Then it is shown that a new class of all-dielectric (all-magnetic), omnidirectional, isotropic, polarization-independent, and transparent 3D media with real positive (>1) permittivity (permeability) can be introduced using T-SUSY. Here is the question: if the material is polarization-dependent (consider an ideal polarizer) that reflects for one polarization and transmits for the corresponding orthogonal polarization, will T-SUSY be reflectionless, for all polarizations? In other words, if n_{T2} gives reflectionless class of materials for the same polarizer at one polarization, will the amount of reflection be 100% for the orthogonal polarization with the same n_{T2} , or n_{T2} needs to be changed to obtain 100% reflection in orthogonal polarization. I think authors need to address this in the paper.

The reviewer proposes a very interesting scenario related to this remarkable feature of T-SUSY, that is, the property of being polarisation independent.

To analyse this particular example, note that an ideal polariser (which completely reflects one polarisation and completely transmits the orthogonal one) can be modelled (for a non-magnetic material) by a time-invariant anisotropic heterogeneous medium characterised by an electric permittivity tensor of the form $\boldsymbol{\epsilon}_{r1}(\mathbf{r}, t) = n_{T1}^2(t)\boldsymbol{\epsilon}_S(\mathbf{r}) = n^2\boldsymbol{\epsilon}_S(\mathbf{r})$. Now consider a reflectionless T-SUSY refractive index partner of $\boldsymbol{\epsilon}_{r1}(\mathbf{r}, t)$ given by $\boldsymbol{\epsilon}_{r2}(\mathbf{r}, t) = n_{T2}^2(t)\boldsymbol{\epsilon}_S(\mathbf{r})$ at a design frequency $\omega = \omega_0$. For simplicity, assume that $n_{T2}(t \rightarrow -\infty) = n_-$ [note that this implies that $n_{T1}(t \rightarrow \infty) = n_{T2}(t \rightarrow \infty) \equiv n_+$ with $n_+ = n_-$, see Supplementary Eq. (2.6)]. As demonstrated in Supplementary Section 1, the electric flux density can be expressed in this case as:

$$\mathbf{D}^{(1,2)}(\mathbf{r}, t) = \psi^{(1,2)}(t)\boldsymbol{\Phi}(\mathbf{r}). \quad (\text{r3.15})$$

This results in two uncoupled wave equations; one of them fully determining the field spatial variation (including the medium response to each field polarisation):

$$\nabla \times \nabla \times (\boldsymbol{\epsilon}_S^{-1}(\mathbf{r})\boldsymbol{\Phi}(\mathbf{r})) - \frac{\omega^2}{c_0^2}n_-^2\boldsymbol{\Phi}(\mathbf{r}) = \mathbf{0}, \quad (\text{r3.16})$$

where ω is the angular frequency of $\psi^{(1,2)}$ at $t \rightarrow -\infty$, and another equation governing the field time dependence:

$$\left(\frac{d^2}{dt^2} + \omega^2 \frac{n_-^2}{n_{T1,2}^2(t)} \right) \psi^{(1,2)}(t) = 0. \quad (\text{r3.17})$$

Now assume that the original time-invariant polariser occupies the region $0 \leq z \leq L$ and that it completely reflects (transmits) the x -polarised (y -polarised) components of a z -propagating wave with an angular frequency ω_0 . For this device, the solution to Eq. (r3.17) is just $\psi^{(1)}(t) = e^{i\omega_0 t}$ so, according to our assumption and to Eq. (r3.15) we have:⁴

$$\mathbf{D}^{(1)}(\mathbf{r}, t) = \psi^{(1)}(t)\boldsymbol{\Phi}(\mathbf{r}) = e^{i\omega_0 t}\boldsymbol{\Phi}(\mathbf{r}) = e^{i\omega_0 t} \begin{cases} A \cos(kz + \varphi_R/2) \hat{x} + B e^{-ikz} \hat{y}, & z < 0 \\ B e^{-i(kz + \varphi_T)} \hat{y}, & z > L \end{cases}, \quad (\text{r3.18})$$

where A and B are complex constants, and where φ_R and φ_T are real constants encoding the phase shift introduced by the polariser for the reflected and transmitted waves, respectively. On the other hand, since n_{T2} is reflectionless, we will have $\psi^{(2)}(t) \sim \psi_+^{(2)}(t) = T_2 e^{i\omega_0 t}$ at $t \rightarrow \infty$ [$|T_2| = |T_1| = 1$, see Supplementary Eq. (2.13)], where the symbol \sim represents an equivalence relation. Hence, after the time modulation (at $t \rightarrow \infty$), the complete solution for the electric flux density in the T-SUSY medium will be:

$$\mathbf{D}^{(2)}(\mathbf{r}, t) \sim \psi_+^{(2)}(t)\boldsymbol{\Phi}(\mathbf{r}) = T_2 e^{i\omega_0 t} \begin{cases} A \cos(kz + \varphi_R/2) \hat{x} + B e^{-ikz} \hat{y}, & z < 0 \\ B e^{-i(kz + \varphi_T)} \hat{y}, & z > L \end{cases}. \quad (\text{r3.19})$$

⁴ The cosine term in Eq. (r3.18) emerges from the linear combination of the incident and reflected x -polarised waves:

$$A' e^{-ikz} + e^{i\varphi_R} e^{ikz} = A' e^{i\frac{\varphi_R}{2}} [e^{-i(kz + \frac{\varphi_R}{2})} + e^{i(kz + \frac{\varphi_R}{2})}] = 2A' e^{i\frac{\varphi_R}{2}} \cos\left(kz + \frac{\varphi_R}{2}\right) \equiv A \cos\left(kz + \frac{\varphi_R}{2}\right)$$

That is, the fields are exactly the same as in the time-independent initial polariser characterised by $\epsilon_{r1}(\mathbf{r}, t)$, except for a possible global phase shift that equally multiplies the incident and reflected waves (i.e., those at $z < 0$) and the transmitted wave (that at $z > L$). Therefore, n_{T2} preserves the spatial scattering properties of $\epsilon_{r1}(\mathbf{r}, t)$ (relation between these waves) for both polarisations, having no influence on the polarisation dependence of the optical system, in this case the ability of completely reflecting one polarisation and fully transmitting the orthogonal one. Consequently, n_{T2} does not need to be changed to obtain 100% reflection in the polarisation blocked by the device.

Following the reviewer's suggestion, we have highlighted this fundamental property of our work in the R version of the paper. Specifically, we have updated Supplementary Eq. (1.9) so as to account for anisotropic media and we have included the previous discussion on the ideal polariser in Section 1 of the Supplementary Information, on page 3. Additionally, we have modified the main text on page 4 as:

"...(e.g., ability of guiding or reflecting/refracting the fields in a specific way for each direction and polarisation; see Supplementary Section 1 for an example involving an ideal polariser, which shows that the spatial response associated with a time-invariant refractive index is preserved by its T-SUSY partner for all polarisations simultaneously)..."

and on page 6 as:

"This allows us to readily generate families of temporal index profiles exhibiting the same scattering intensity as another medium (which can have any spatial variation and polarisation response)..."

Minor revisions and typos detected by the authors (for Associate Editor and Referees)

R Paper and R Supplementary Information:

- Figure 4(c) of the R paper has been updated by including the label “Index perturbation”
- Data and code availability statements included on page 14 of the R1 paper
- For a better visualisation, Supplementary Fig.3.1(c) has been updated by including an additional inset depicting the data for $\Omega/\omega_0^2 \in (1,1.01]$
- We have amended Supplementary Section 7, as we noted that, while the acoustic equation included in the original version of this document is valid for a mass density with any time dependence (the case discussed in the manuscript), it only accounts for a slowly-varying bulk modulus. The modified equation in the R version of the Supplementary Information is completely general. The prescription for the implementation of T-SUSY media via a time-varying mass density is not affected by this modification. In this general scenario, it is also possible to synthesize T-SUSY modulations based on a time-varying bulk modulus (or on both a time-varying density and bulk modulus), as described in the R Supplementary Information.
- We have updated the figures and equations of the R Supplementary Information according to the journal’s style.

REVIEWERS' COMMENTS:

Reviewer #2 (Remarks to the Author):

In a previous report I stated that, although a fascinating idea, I did not think that this work showed any realistic potential experimental realisation.

In this revised version the authors have done further work in this direction, and in particular Table 1 has now been added which shows the time scales associated with some potential platforms. This goes a long way to answering my previous objection.

However, because the scope of the paper is now broader the text is now slightly clunky in places. One can definitely tell that the acoustics/elasticity results were added later. For instance, the title could be changed, and some of the text edited to emphasise that this is now a more universal theory.

Once these suggested cosmetic changes have been made I think this is suitable for publication.

Reviewer #3 (Remarks to the Author):

I would like to thank the authors for addressig my comments. All the comments have been addressed very satisfactorily and the appropriate changes have been applied in the revised manuscript. I believe that the paper is now ready for publication in Nature Communications.

Feedback from Reviewer #2 Comments

In a previous report I stated that, although a fascinating idea, I did not think that this work showed any realistic potential experimental realisation. In this revised version the authors have done further work in this direction, and in particular Table 1 has now been added which shows the time scales associated with some potential platforms. This goes a long way to answering my previous objection. However, because the scope of the paper is now broader, the text is now slightly clunky in places. One can definitely tell that the acoustics/elasticity results were added later. For instance, the title could be changed, and some of the text edited to emphasise that this is now a more universal theory. Once these suggested cosmetic changes have been made I think this is suitable for publication.

We are pleased to receive such positive feedback from the referee on our work and the last improvements we made.

We also thank the reviewer for her/his suggestion, on which we completely agree. Following this recommendation, we have harmonized the manuscript taking into account the more universal character of the theory.

In particular, we have changed the title as:

“Supersymmetry in the time domain and its applications in optics”

We believe that this slight change captures the fact that the theory is more general, and at the same time highlights the fact that the work is focused on the optical part.

Additionally, we have rewritten the abstract along the same line (note that we have also extended the first background abstract sentences, following the editor’s suggestion). The new version reads:

“Supersymmetry is a conjectured symmetry between fermions and bosons aiming at solving fundamental questions in string and quantum field theory. Its subsequent application to quantum mechanics led to a ground-breaking analysis and design machinery, later fruitfully extrapolated to photonics. In all cases, the algebraic transformations of quantum-mechanical supersymmetry were conceived in the space realm. Here, we demonstrate that Maxwell’s equations, as well as the acoustic and elastic wave equations, also possess an underlying supersymmetry in the time domain. We explore the consequences of this property in the field of optics, obtaining a simple analytic relation between the scattering coefficients of numerous time-varying systems, and uncovering a wide class of reflectionless, three-dimensional, all-dielectric, isotropic, omnidirectional, polarisation-independent, non-complex media. Temporal supersymmetry is also shown to arise in dispersive media supporting temporal bound states, which allows engineering their momentum spectra and dispersive properties. These unprecedented features enable the creation of novel reconfigurable devices, including invisible materials, frequency shifters, isolators, and pulse-shape transformers.”

Finally, we have performed the following modifications in the Introduction, Discussion and References sections:

1. Introduction, Page 3, last two paragraphs:

“Remarkably, however, the fact that the temporal derivative in the electromagnetic, **acoustic, and elastic wave equations** is of second order may enable a temporal version of SUSYQM, which has been overlooked so far. This would extend the foundations and unique properties of SUSYQM to the time domain, adding an unprecedented degree of understanding and control **over time-varying systems in**

various fields of physics, and opening the door to a myriad of new applications. Actually, time-varying optical systems are becoming crucial in a broad range of scenarios, including optical modulation [16], isolation and non-reciprocity [17,18], all-optical signal processing [19,20], quantum information [21], and reconfigurable photonics [22,23]. Likewise, temporal modulations enable new possibilities for the manipulation of sound and mechanical oscillations [24-26].

Here, it is shown that Maxwell's equations indeed possess an underlying time-domain supersymmetry (T-SUSY) for any non-dispersive optical system characterised by a refractive index of the form:

$$n(\mathbf{r}, t) = n_S(\mathbf{r})n_T(t) = \sqrt{\varepsilon_S(\mathbf{r})\mu_S(\mathbf{r})}\sqrt{\varepsilon_T(t)\mu_T(t)}, \quad (2)$$

where $\varepsilon_T(\mathbf{r}, t) = \varepsilon_S(\mathbf{r})\varepsilon_T(t)$ is the medium relative permittivity and $\mu_T(\mathbf{r}, t) = \mu_S(\mathbf{r})\mu_T(t)$ its relative permeability, with similar results for acoustic and elastic waves (T-SUSY can also be found in dispersive systems, as discussed below, and in anisotropic and nonlocal media, as discussed in Supplementary Note 1). In the following, the T-SUSY formalism is developed for the field of optics, analysing both the continuous and discrete spectrum cases, and illustrating its potential through different applications (sketched in Fig. 1). Finally, the extension of T-SUSY to transmission line theory, acoustics and elasticity is discussed, assessing the experimental opportunities offered by current technological platforms."

2. Page 12, first paragraph of the Discussion section:

"Additionally, as outlined above and as shown in detail in Supplementary Notes 7 and 8, both sound and elastic waves satisfy a temporal Helmholtz equation formally equal to equation (3). Hence, T-SUSY can be directly transferred to these fields of physics."

3. Reference list. We have added the following reference:

Fleury R., Khanikaev A. B. & Alù A. Floquet topological insulators for sound. Nature Commun. 7, 11744 (2016).

Feedback from Reviewer #3 Comments

I would like to thank the authors for addressing my comments. All the comments have been addressed very satisfactorily and the appropriate changes have been applied in the revised manuscript. I believe that the paper is now ready for publication in Nature Communications.

We sincerely appreciate the referee's positive comments and assessment of our work.